# Mouse *Tmem135* mutation reveals a mechanism involving mitochondrial dynamics that leads to age-dependent retinal pathologies

Wei-Hua Lee[1], Hitoshi Higuchi[1†], Sakae Ikeda[1,2], Erica L Macke[1], Tetsuya Takimoto[1], Bikash R Pattnaik[2,3], Che Liu[4,5], Li-Fang Chu[6], Sandra M Siepka[7‡], Kathleen J Krentz[8], C Dustin Rubinstein[9], Robert F Kalejta[4,5], James A Thomson[6], Robert F Mullins[10], Joseph S Takahashi[11], Lawrence H Pinto[7], Akihiro Ikeda[1,2]*

[1]Department of Medical Genetics, University of Wisconsin-Madison, Madison, United States; [2]McPherson Eye Research Institute, University of Wisconsin-Madison, Madison, United States; [3]Department of Pediatrics, University of Wisconsin-Madison, Madison, United States; [4]Institute for Molecular Virology, University of Wisconsin-Madison, Madison, United States; [5]McArdle Laboratory for Cancer Research, University of Wisconsin-Madison, Madison, United States; [6]Morgridge Institute for Research, Madison, United States; [7]Department of Neurobiology, Northwestern University, Evanston, United States; [8]Transgenic Mouse Facility, Biotechnology Center, University of Wisconsin-Madison, Madison, United States; [9]Translational Genomics Facility, Biotechnology Center, University of Wisconsin-Madison, Madison, United States; [10]Department of Ophthalmology and Visual, University of Iowa, Iowa City, United States; [11]Department of Neuroscience, Howard Hughes Medical Institute, University of Texas Southwestern Medical Center, Dallas, United States

*For correspondence: aikeda@wisc.edu

Present address: [†]Department of Dental Anesthesiology, Okayama University Hospital, Okayama, Japan; [‡]The Chemistry of Life Processes Institute, Northwestern University, Evanston, United States

**Abstract** While the aging process is central to the pathogenesis of age-dependent diseases, it is poorly understood at the molecular level. We identified a mouse mutant with accelerated aging in the retina as well as pathologies observed in age-dependent retinal diseases, suggesting that the responsible gene regulates retinal aging, and its impairment results in age-dependent disease. We determined that a mutation in the transmembrane 135 (*Tmem135*) is responsible for these phenotypes. We observed localization of TMEM135 on mitochondria, and imbalance of mitochondrial fission and fusion in mutant *Tmem135* as well as *Tmem135* overexpressing cells, indicating that TMEM135 is involved in the regulation of mitochondrial dynamics. Additionally, mutant retina showed higher sensitivity to oxidative stress. These results suggest that the regulation of mitochondrial dynamics through TMEM135 is critical for protection from environmental stress and controlling the progression of retinal aging. Our study identified TMEM135 as a critical link between aging and age-dependent diseases.

## Introduction

One explanation for why age-dependent diseases manifest themselves in an age-dependent manner is that disease-causing mechanisms interact with age-dependent cellular changes that normally occur

**eLife digest** Older people have an increased risk of developing many diseases, such as diabetes and age-related macular degeneration (which is often shortened to AMD). This suggests that changes that occur during normal aging may some how be linked to how such diseases develop. However, the molecular mechanisms responsible for these links are not clear.

AMD causes damage to the retina of the eye, which can lead to visual loss in older people. To investigate the link between aging and age-dependent diseases, Lee et al. used mutant mice whose retina of the eye ages more quickly than normal mice and are prone to developing an eye condition that is similar to AMD. The experiments show that these mice have a mutation in a gene called *Tmem135* that is responsible for these visual problems. *Tmem135* regulates the size of cell compartments called mitochondria, which produce energy for the cell. This affects the ability of the mitochondria to work properly and makes the cells more sensitive to environmental stress, which in turn makes the retina age more quickly.

The findings of Lee et al. show that *Tmem135* is a critical link between aging and an AMD-like condition in mice. Furthermore, the experiments suggest that defects in mitochondria may accelerate the normal pace of aging and lead to AMD and other age-dependent diseases. Further studies are needed to find out exactly what role *Tmem135* plays in mitochondria and whether it also contributes to the aging of other parts of the body.

in aging. The common phenomena observed in both aging and age-dependent diseases may provide clues to this interaction. For example, one of the major age-dependent changes that generally occur in the tissue is accumulation of damages caused by oxidative stress (*Harman, 1956*, *1972a*, *1972b*). As by-products of normal cellular respiration, reactive oxygen species (ROS) are constantly generated in cells mainly in the mitochondria. When cellular production of ROS overwhelms its antioxidant capacity (state referred to as 'oxidative stress'), ROS damages cellular macromolecules such as lipids, protein, and DNA/RNA. In the course of aging, such damages caused by ROS are thought to accumulate and contribute to the development of age-dependent tissue dysfunctions. Increase in oxidative damage has been observed in a number of age-dependent diseases as well, and its involvement in the pathogenesis of these diseases has been widely suggested (*Davies, 1995*). Related to the oxidative damage, another phenomenon that is observed in both aging and age-dependent diseases is the decline in mitochondrial function (*Lenaz, 1998*). Mitochondria are the organelle that consumes over 90% of cellular oxygen and generates ROS (*Harman, 1981*; *Murphy, 2009*). Due to its proximity to the site of ROS generation, mitochondrial components are particularly susceptible to ROS-mediated oxidative damage (*Cadenas and Davies, 2000*). Prolonged exposure to ROS during aging is thought to result in mitochondrial dysfunctions and significantly contribute to the development of pathologies associated with aging. There is also strong evidence that mitochondrial dysfunction occurs early and acts causally in the pathogenesis of age-dependent neurodegenerative diseases (*Lin and Beal, 2006*). While these phenomena indicate some of the common aspects between aging and age-dependent diseases, the mechanisms linking these two processes have not been elucidated at the molecular level.

Given the complexity in both the aging process and age-dependent diseases, as well as countless variables (including genetic an environmental variables) that exist among human population, it is extremely challenging to study the mechanisms underlying the aging process and how they relate to the disease-causing mechanism in humans. An animal model that shows accelerated aging as well as age-dependent disease symptoms could provide a useful experimental system for this purpose. Furthermore, a forward genetics approach starting with an animal model with these symptoms offers a potential of identifying a responsible gene that is not previously known to be associated with the aging process nor age-dependent diseases. We isolated an N-ethy-N-nitrosourea (ENU)-induced mutant mouse line, *FUN025*, that exhibits age-dependent retinal abnormalities with a trajectory similar to that found with retinal aging observed in wild-type (WT) mice (*Higuchi et al., 2015*) but with an early onset and faster progression. In addition, we found that the *FUN025* mutation leads to pathologies observed in age-dependent retinal diseases such as age-related macular degeneration

(AMD). These phenotypes in *FUN025* mice suggest that the responsible gene is involved in regulating the rate of aging in the retina, and that its impairment leads to development of age-dependent disease. In this study, we identify a gene mutation that is responsible for retinal abnormalities in *FUN025* mice and characterize the novel molecular functions of this gene/protein associated with regulation of mitochondria as well as sensitivity to oxidative stress. Our findings reveal a molecular link between the aging process and age-dependent diseases, and a molecular mechanisms leading to age-dependent disease pathologies.

## Results

### Early-onset and accelerated progression of aging-associated changes in the *FUN025* retina

*FUN025* mice were isolated through fundus examination in an ENU mouse mutagenesis project (*Pinto et al., 2004*; *Vitaterna et al., 2006*), and were found to exhibit retinal abnormalities similar to those observed in aged WT mice (*Higuchi et al., 2015*). It has been shown that, in the WT retina, age-dependent abnormalities including retinal degeneration, increased number of ectopic synapses and increased retinal stress all start from the peripheral retina by 8 months of age on an aging-susceptible A/J background but later on a less susceptible B6 background, which progress to the central retina with age (*Higuchi et al., 2015*). Histological analysis of homozygous *FUN025* mutant and C57BL/6J (B6) WT retina at two and seven months of age revealed that a decrease in the ONLT Index [the thickness of outer nuclear layer (ONL) normalized by the thickness of inner nuclear layer (INL)] in *FUN025* mice compared to WT mice is observed by two months of age and becomes more pronounced by seven months of age, indicating progressive loss of photoreceptor cells in the *FUN025* retina (*Figure 1A,B*). Immunohistochemical analysis demonstrated that, while most of the presynaptic photoreceptor terminals (PSD95, green) line up in the outer plexiform layer (OPL) in close opposition to the bipolar cell postsynaptic structures (PKC, red) in the WT retina, ectopic localization of presynaptic terminals and abnormal extension of bipolar cell dendrites into the ONL were observed in the *FUN025* retina (*Figure 1C,D*). Quantification of ectopic synapses indicated that a significantly increased number of ectopic synapses are observed in the peripheral retina of *FUN025* mice by two months of age, and in the central retina by seven months of age, indicating that this phenotype progresses from the peripheral to the central retina (*Figure 1D*). We also observed that the sign of retinal stress, up-regulation of glial fibrillary acidic protein (GFAP) (*Higuchi et al., 2015*; *Lewis and Fisher, 2003*), was significantly increased in the peripheral retina of *FUN025* mice compared to the WT mice by two months of age, which later progresses toward the central retina (*Figure 1E*). Thus, *FUN025* retina exhibits age-dependent retinal abnormalities with a trajectory similar to that found with retinal aging observed in WT mice (*Higuchi et al., 2015*) but with an early onset and faster progression.

### *FUN025* mice exhibit age-dependent disease pathologies

We further investigated whether the early-onset and accelerated aging process in the *FUN025* retina is accompanied by pathologies observed in age-dependent diseases. Punctate light deposits were found in the fundus photography of the eyes from *FUN025* mice (*Figure 2A*), which may be due to the accumulation of autofluorescent cells and aggregates observed between photoreceptors and retinal pigment epithelium (RPE) (*Figure 2B*). These cells/aggregates and RPE exhibit autofluorescence detected using a DPS 561 laser (*Figure 2B,C*), which resembles that of lipofuscin, protein and lipid rich aggregates known to accumulate in aging tissues and suggested to be involved in age-dependent diseases such as AMD and Alzheimer's disease (*Giaccone et al., 2011*; *Nowotny et al., 2014*). The hyper-autofluorescent aggregates (*Figure 2B*, arrowhead) resemble subretinal drusenoid deposits (SDD) (*Rudolf et al., 2008*), a type of extracellular lesion between the photoreceptors and RPE often observed in AMD patients (*Curcio et al., 2013*; *Zweifel et al., 2010*). To determine the cell type of these autofluorescent cells, immunostaining was performed in 7-month old retinal sections. The subretinal autofluorescent cells were positive for a microglia marker, Iba1, and a macrophage marker, F4/80, in the *FUN025* retina (*Figure 2C*). The Iba1[+] cells near the apical surface of the RPE have very few processes and show immunoreactivity to F4/80, suggesting that they have transformed from microglia to macrophages (*Figure 2C*). These data also suggest that the RPE layer

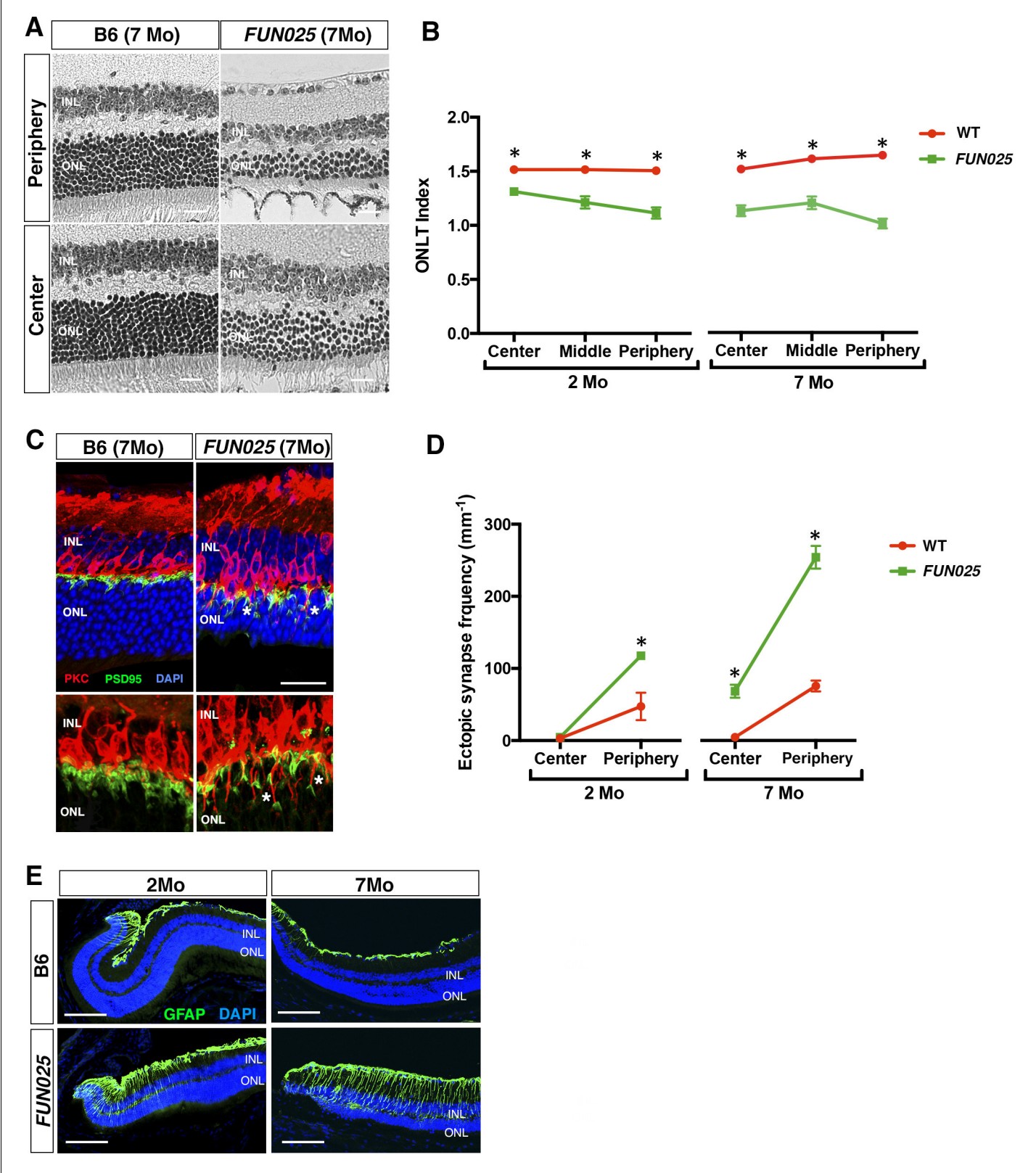

**Figure 1.** Age-dependent retinal abnormalities in *FUN025* mice. (**A–B**) A significant decrease of the ONLT index occurred by two months of age in *FUN025* retina. Mo = months. Data from *n* = 10 WT (2 Mo), *n* = 4 *FUN025* (2 Mo), *n* = 20 WT (7 Mo), *n* = 8 *FUN025* (7 Mo) mice. Scale bar = 20 μm. (**C–D**) Ectopic synapses were observed as bipolar cell neurites (PKC, red) and photoreceptor synaptic terminals (PSD95, green) extending into the ONL

*Figure 1 continued on next page*

*Figure 1 continued*

indicated by asterisks (**C**). Scale bar = 10 μm. Significant increase of ectopic synapses were found earlier in the peripheral retina, and later in the central retina of *FUN025* compared to WT mice. Data for central retina from *n* = 3 WT (2 Mo), *n* = 3 *FUN025* (2 Mo), *n* = 3 WT (7 Mo), *n* = 3 *FUN025* (7 Mo) mice; data for peripheral retina from *n* = 5 WT (2 Mo), *n* = 6 *FUN025* (2 Mo), *n* = 6 WT (7 Mo), *n* = 6 *FUN025* (7 Mo) mice. (**E**) GFAP (green) upregulation was progressively observed in the *FUN025* retina. ONL: outer nuclear layer. INL: inner nuclear layer. Outer nuclear layer thickness (ONLT) index = ONL thickness/INL thickness. *p<0.05, Student's *t*-test. All data are mean ± s.e.m. Scale bar = 50 μm.

is the origin of stress, toward which the microglia migrate (*Jonas et al., 2012*; *Mcglade-Mcculloh et al., 1989*). Long-term, low-grade inflammation (innate immunity) has been widely

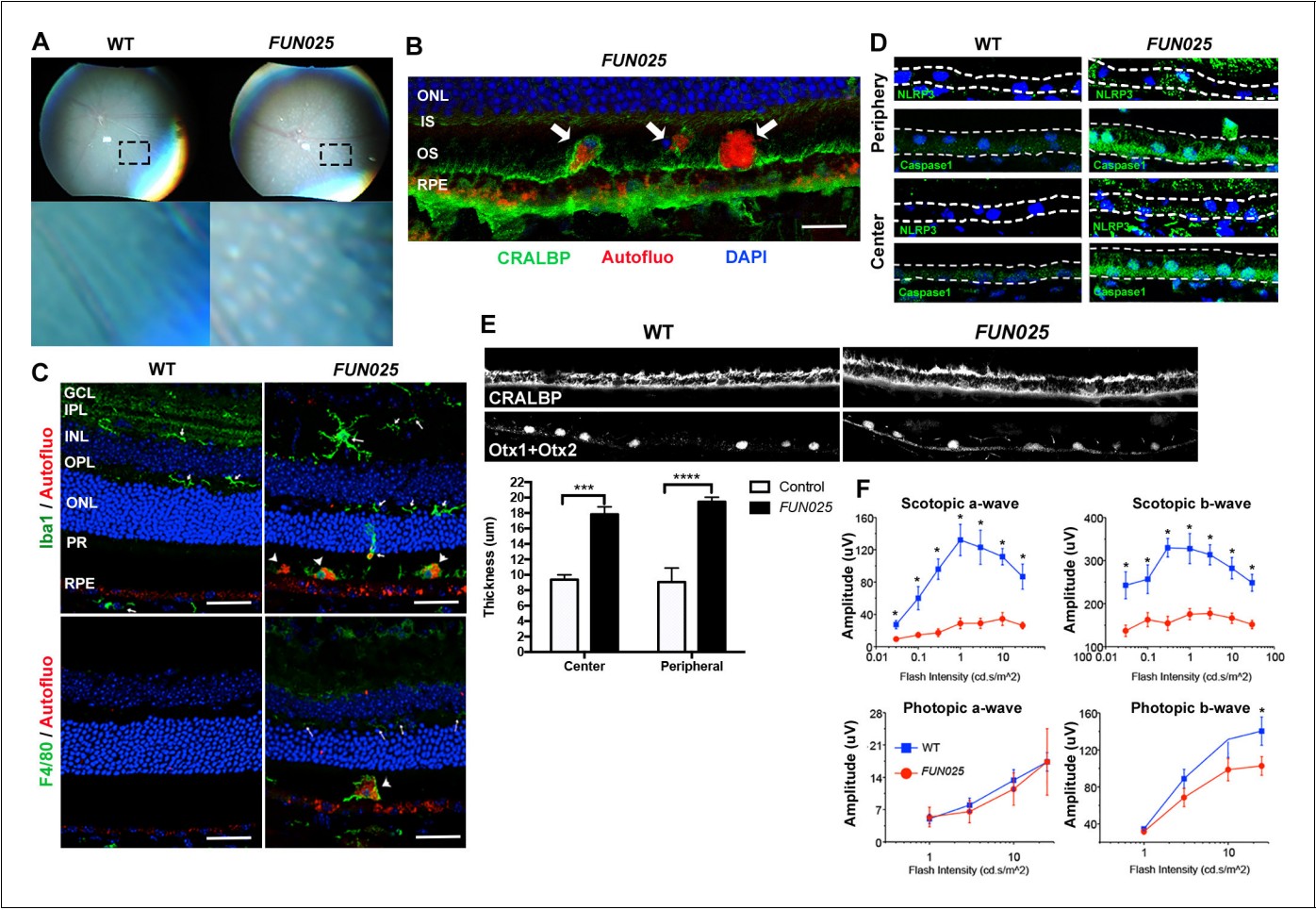

**Figure 2.** *FUN025* mice show AMD-like pathologies. (**A**) Punctate light deposits were found in the fundus photography of the eyes from WT and *FUN025* mice. (**B**) Autofluorescent cells/aggregates (indicated by arrows) and lipofuscin-like autofluorescence were observed in proximity to the apical surface of the RPE in *FUN025* mice. Scale bar = 60 μm. (**C**) Iba1 (microglia/ macrophage marker) and F4/80 (macrophage marker) positive cells were found in the *FUN025* retina at seven months, whereas very few Iba1 positive cells were found in the WT retina at seven months. Scale bar = 20 μm. (**D**) Signals for inflammasome markers, NLRP3 and caspase1, increased in the RPE in both peripheral and central retina from *FUN025* mice compared to WT control. (**E**) At seven months of age, the RPE (highlighted by CRALBP staining) thickness is significantly increased in both central and peripheral retina of *FUN025* mice compared to control mice. The RPE nuclei were highlighted with Otx1+Otx2. Data from *n* = 4 mice per genotype. (**F**) Transcorneal electroretinograms (ERG) recordings from seven-month-old *FUN025* mice and their WT littermates. Both scotopic (dark-adapted) ERG a- and b-waves from the rod pathway were markedly reduced in *FUN025* mice. A reduction was also observed in photopic (light-adapted) ERG b-wave from the cone pathway with higher flash intensity, while no difference between *FUN025* and WT was observed in the photopic a-wave, majority of which is postreceptoral in origin. Data from *n* = 5 mice per genotype. *p<0.05, two-way analysis of variance (ANOVA). All data are mean ± standard error of the mean (s.e.m.).

postulated as a part of the aging process, and is also enhanced in many age-dependent diseases (*Salminen et al., 2012*). To test if innate immunity is increased in *FUN025* mice, immunostaining was performed with an inflammasome marker, NLR Family, Pyrin Domain Containing 3 (NLRP3) and a functional caspase-1 subunit marker, caspase-1 p10 on retinal sections from seven-month old *FUN025* and WT mice. Immunoreactivity to both makers was increased in the neural retina as well as RPE of *FUN025* mice compared to WT mice (*Figure 2D*), indicating elevated innate immunity (*Franchi et al., 2009*). In addition, by seven months, the RPE (highlighted by anti-CRALBP antibody) is uniformly thickened in *FUN025* mice compared to that of controls while each RPE cell is still intact (the nuclei of RPE highlighted by anti-Otx1+Otx2 antibody) (*Figure 2E*). Similar increases in innate immunity markers (*Tarallo et al., 2012*) and thickness of RPE (*Acton et al., 2012*; *Karampelas et al., 2013*; *Zhao and Vollrath, 2011*) have been observed in AMD patients. To determine if early-onset of retinal aging as well as retinal pathologies observed in *FUN025* mice affect the ability of the retina to respond to light, we performed transcorneal electroretinograms (ERG) recordings from seven-month-old *FUN025* mice and their WT littermates. ERG data revealed significantly reduced scotopic (dark-adapted) a-wave and b-wave from the rod pathway, and a modest reduction in photopic (light-adapted) b-wave from the cone pathway in *FUN025* mice compared to WT controls (*Figure 2F*) indicating impaired visual function in *FUN025* mice. In conclusion, *FUN025* mice provide a mouse model with accelerated aging process and age-dependent disease pathologies in the retina.

## Identification of a *Tmem135* mutation in *FUN025* mice

To identify the causative gene/mutation for retinal phenotypes in *FUN025* mice, we performed a genome-wide linkage analysis. Genetic mapping of the *FUN025* mutation was performed using F2 intercrosses (C57BL6-*FUN025* x C57BR/cdJ and C57BL6-*FUN025* x 129S1/SvImJ) and genetic markers to distinguish between the alleles of *FUN025* (C57BL/6J) and C57BR/cdJ or 129S1/SvImJ (*Figure 3A*). The F2 mice were phenotyped by fundus photography and histological analysis at 3 months. Genotype and phenotype data from 70 F2 mice generated a minimal genetic region flanked by SNPOlf294T>C (93764448) and Rs31011252, containing 25 genes. Using a SureSelectXT custom library (Agilent Technologies), we performed a sequence capture array on *FUN025* genomic DNA followed by paired end sequencing. Standard bioinformatic analyses of our sequencing data revealed a point mutation (T>C) in the splice-donor site adjacent to exon 12 of *Tmem135* in *FUN025* mice (*Figure 3B,C*). The consensus sequence of mouse splice donor sites depicts the necessity of the GT sequence at positions 1 and 2 downstream of the exon boundary for the functionality of the site (*Carmel et al., 2004*). The mutation disrupts the splice donor site (*Figure 3C*), resulting in skipping of exon 12 and a frame shift creating an early stop codon in *FUN025* mice (*Figure 3D,F*). The probability of forming transmembrane helices predicted by the program TMHMM (v. 1.0) (http://www.cbs.dtu.dk/) suggested that WT TMEM135 contains five transmembrane helices (*Figure 3G* and *Figure 3—figure supplement 1*), while the 4th and 5th transmembrane helices are abolished and the orientation of the remaining 3 transmembrane helices in the membrane is reversed in mutant TMEM135 compared to WT TMEM135 (*Figure 3G* and *Figure 3—figure supplement 1*). The c-terminal region of the mutant TMEM135 is also shorter due to the early stop codon (*Figure 3E,G*). Thus, the *FUN025* mutation in *Tmem135* which results in the shorter c-terminal region with amino acid sequence changes, predicted loss of two transmembrane helices and reversed orientation within the membrane likely impairs the normal functions of the TMEM135 protein. The TMEM135 antibody recognizing the N-terminus of TMEM135 protein detected the mutant TMEM135 protein in the *FUN025* brains as well as the WT TMEM135 protein in WT brains by western blot analysis (*Figure 3H*).

To confirm that this mutation, rather than any other unknown mutations that occurred in the ENU-induced *FUN025* mutant line, is responsible for the retinal phenotypes in *FUN025* mice, a complementation test was performed. We introduced the same point mutation (T > C) as observed in *FUN025* mice in the intron 12 of *Tmem135* (Chr7:96,296,478) in C57BL/6J mice using the CRISPR/Cas9 system (T > C mice). The T > C heterozygous mice were crossed with the *FUN025* homozygous mice to produce F1 (T > C/*FUN025* compound heterozygous) mice, which were analyzed for retinal phenotypes. The F1 mice exhibited retinal phenotypes similar to *FUN025* homozygous mice, indicating non-complementation (*Figure 3—figure supplement 2*). This result demonstrates that the point

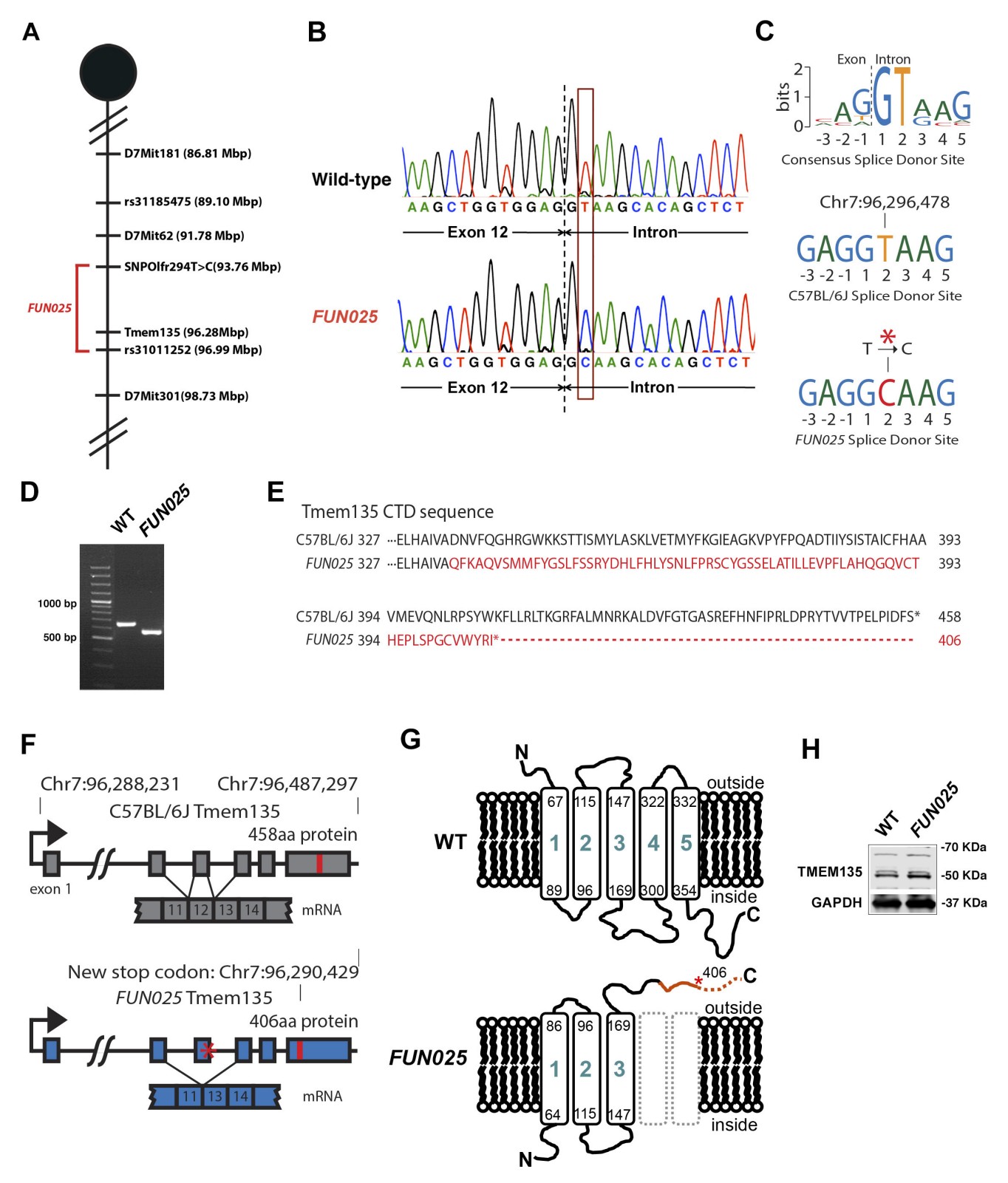

**Figure 3.** Identification of a *Tmem135* mutation in *FUN025* mice. (**A**) Minimal genetic region of *FUN025* on chromosome 7 determined by genetic mapping. (**B**) A point mutation (T > C) in the splice-donor site adjacent to exon 12 of *Tmem135* in *FUN025* mice. (**C**) The consensus sequence of mouse splice donor sites, depicting the necessity of the GT sequence at positions 1 and 2 downstream of the exon boundary for the functionality of the site. The C57BL/6J and *FUN025* sequences are shown below, demonstrating the disrupted site in *FUN025* mice. (**D**) RT-PCR spanning exon 12 and

*Figure 3 continued on next page*

*Figure 3 continued*

sequencing the product revealed the absence of this exon in the *FUN025* retina. (E) Amino acid sequences of the C-terminus of TMEM135 in C57BL/6J and *FUN025* mice. The WT protein is 458 amino acids long, whereas the truncated mutant protein is 406. The change in amino acid sequence is highlighted in red. (F) Consequences of the mutation in the genomic sequence of *Tmem135*. The mutation adjacent to exon 12 of *Tmem135* (red star) results in a non-functional splice donor site, causing skipping of exon 12. This results in a frameshift and an early stop codon (chr7: 96,290,429, NCBI build 37). Locations of the stop codons are highlighted in red. (G) A predicted structure of TMEM135 having five transmembrane domains. The *FUN025* mutation is predicted to result in a protein with only three transmembrane domains, whose orientation in the membrane is reversed. The rest of the c-terminal region is absent due to the early stop codon (asterisk). (H) Western blot for TMEM135 in WT and *FUN025* brains. GAPDH was used as a loading control.

The following figure supplements are available for figure 3:

**Figure supplement 1.** Identification of the causative gene, *Tmem135*, for the *FUN025* mutation.

**Figure supplement 2.** A T > C mutation in *Tmem135* fails to complement *FUN025*.

mutation (T > C) in *Tmem135* is indeed the *FUN025* causative mutation. Therefore, we now designate homozygous *FUN025* mice as *Tmem135*<sup>FUN025/FUN025</sup>.

## Localization of TMEM135

TMEM135 protein was suggested to be involved in fat storage and the regulation of longevity in *C. elegans* (*Exil et al., 2010*), but the function of TMEM135 is not yet clearly characterized. In order to elucidate the mechanistic role of TMEM135, we characterized the localization of the TMEM135 protein in cultured cells and mouse retina in vivo. Primary WT mouse fibroblast cells (MFs) were co-transfected with the green fluorescent protein (GFP)-tagged vector containing *Tmem135* and DsRed2-tagged mitochondria vector (pDsRed2-Mito Vector, Clontech, Mountain View, CA). Immunofluorescence data showed that the GFP-TMEM135 protein displays an intracellular vesicular expression pattern in the cytoplasm, and a proportion is found in punctate structures on mitochondria (*Figure 4A*). Colocalization of TMEM135 to mitochondria was further confirmed in WT MFs using an anti-TMEM135 antibody and an AcGFP1-tagged mitochondria vector (pAcGFP1-Mito Vector, Clontech, Mountain View, CA) (*Figure 4A*) as well as in WT MFs with anti-TMEM135 antibody and a red-fluorescent dye that stains mitochondria (MitoTracker Red CMXRos) (*Figure 4B*, upper panel). Notably, the proportion of TMEM135 signals tend to distribute to small foci along the surface of mitochondria, at mitochondrial constriction sites, and at the tips of individual mitochondria (*Figure 4A,B*), and there appears to be less colocolization of TMEM135 on mitochondria in *FUN025* MFs (*Figure 4B*, lower panel). Immunoelectron microscopy revealed that transfected GFP-TMEM135 localizes on the surface of mitochondria (*Figure 4C*). Furthermore, colocalization of TMEM135 to mitochondria was observed in monkey kidney fibroblast-like cells (Cos-7) (*Figure 4D*) and mouse primary hippocampal neurons (*Figure 4E*). The mitochondria fraction isolated from mouse brains showed TMEM135 signals by immunoblotting, also indicating that TMEM135 is associated with mitochondria (*Figure 4F*). In the retina from WT and *FUN025* mice, stronger TMEM135 signals were detected in the ganglion cell layer (GCL), inner plexiform layer (IPL), outer plexiform layer (OPL), inner segments of photoreceptor cells, and RPE, which colocalized with mitochondria labeled with anti-TOMM20 antibody (*Figure 4G*). The colocalization of TMEM135 with mitochondria was also observed in the primary mouse RPE cell culture (*Figure 4H*). Similar to the observation in fibroblasts (*Figure 4B*), less colocalization of TMEM135 with mitochondria was observed in *FUN025* RPE cells compared to WT RPE cells (*Figure 4H*). In conclusion, a proportion of TMEM135 is strongly associated with mitochondria in vivo, although other intracellular organelles and small vesicles may be also associated with TMEM135.

## TMEM135 plays a role in mitochondrial dynamics

We next investigated what roles TMEM135 might play in mitochondria. TMEM135 punctate structures were often observed at mitochondrial constriction sites and at the tips of individual mitochondria. This unique localization pattern suggests a role of TMEM135 in the regulation of mitochondrial morphology. We isolated MFs from WT, *Tmem135*<sup>FUN025/FUN025</sup> and transgenic mice

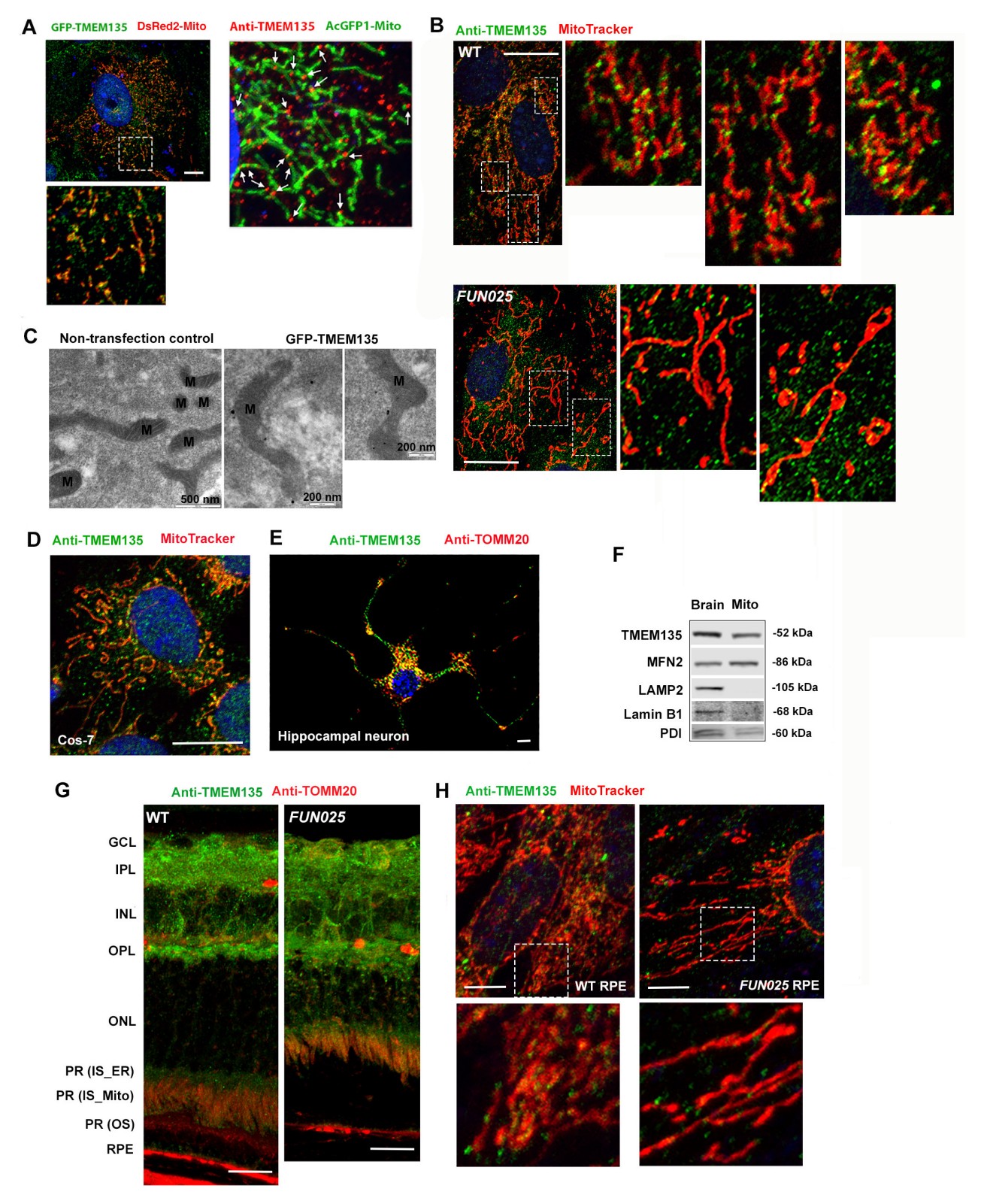

**Figure 4.** Localization of TMEM135 to the mitochondria. (A) Mitochondrial localization of TMEM135 in MFs co-transfected with GFP tagged TMEM135 vector (green) and DsRed2 tagged mitochondria vector (red). GFP-TMEM135 signals were detected as puncta to the mitochondria as well as in the cytoplasm. Colocalization of TMEM135 and mitochondria in MFs transfected with AcGFP1 tagged mitochondria vector (green) and immunostained with anti-TMEM135 antibody (red). Scale bar = 10 μm. (B) Colocalization of TMEM135 (anti-TMEM135 antibody, green) and mitochondira (MitoTracker, red)

*Figure 4 continued on next page*

*Figure 4 continued*

in wild-type and *FUN025* mouse fibroblasts. Scale bar = 10 μm. (C) Immuno-EM revealed localization of GFP-tagged TMEM135 to the mitochondria. (D–E) Colocalization of TMEM135 (anti-TMEM135 antibody, green) and mitochondria (MitoTracker and TOMM20, red) in Cos-7 cells and primary mouse hippocampal neuron. Scale bar = 10 μm. (F) The mitochondrial fraction isolated from the WT mouse brain show TMEM135 signals by immunoblotting. Following proteins were used as organelle markers: MFN2–mitochondria; LAMP2–lysosome; Lamin B1– nucleus; PDI– endoplasmic reticulum (ER). (G) Strong TMEM135 signals (green) in GCL, IPL, OPL, inner segments of photoreceptor cells, and RPE from wild-type and *FUN025* mouse retina. Throughout the retina, TMEM135 is colocalized with mitochondria (anti-TOMM20 antibody, red). Scale bar = 10 μm. (H) Colocalization of TMEM135 (anti-TMEM135 antibody, green) and mitochondria (MitoTracker, red) in wild-type and *FUN025* primary mouse RPE cell culture. Scale bar = 5 μm.

overexpressing WT *Tmem135* (Tg-Tmem135) (*Figure 5—figure supplement 1A–B*), and stained the mitochondria with MitoTracker Red. Compared to WT cells with mitochondria of all different sizes and shapes, *Tmem135*$^{FUN025/FUN025}$ cells showed over-fused mitochondrial networks whereas Tg-Tmem135 cells exhibited over-fragmented mitochondrial networks (*Figure 5A*). We used a morphology scoring assay (*Loson et al., 2013*) in which each cell was categorized as having fragmented, tubular, elongated or aggregated mitochondria. Among *Tmem135*$^{FUN025/FUN025}$ MFs, more cells were found to have elongated mitochondria relative to cells isolated from WT mice, indicating that more fusion occurs compared to fission in *Tmem135*$^{FUN025/FUN025}$ cells (*Figure 5B*). In contrast, among Tg-Tmem135 MFs, more cells were found to have fragmented mitochondria relative to WT cells, suggesting that more fission than fusion takes place in Tg-Tmem135 cells (*Figure 5B*). Next, we compared the number and size of mitochondria between *Tmem135*$^{FUN025/FUN02}$, Tg-Tmem135 and WT MFs (*Figure 5C–E*). The size of mitochondria increases and its number decreases in *Tmem135*$^{FUN025/FUN025}$ cells indicating that more mitochondrial fusion occurs than fission leading to elongated mitochondria. In contrast, the size and mass of mitochondria decrease in Tg-Tmem135 cells indicating that more fission than fusion takes place producing over-fragmented mitochondria. Observation that mitochondrial size increases in *Tmem135*$^{FUN025/FUN025}$ cells was further confirmed by knocking down *Tmem135* using RNA interference (RNAi). Knocking down 68% of *Tmem135* RNA in WT MFs using a short interfering RNA (siRNA) against *Tmem135* resulted in a significant increase in mitochondrial size compared to WT MFs treated with scrambled siRNA (*Figure 5F*). We next analyzed the retinal mitochondrial morphology in the RPE as well as inner segments of photoreceptor cells from 12-month-old WT and *FUN025* mice using electron microscopy (*Figure 5G*). We observed enlarged mitochondria in both the RPE and inner segments of photoreceptor cells from *FUN025* mice compared to those from WT mice (*Figure 5G,H*). Taken together, these results indicate that TMEM135 is involved in the regulation of the balance between mitochondrial fission and fusion (mitochondrial dynamics).

## Does TMEM135 promote fission or inhibit fusion?

Two possible mechanisms underlie the mitochondrial morphological changes in *Tmem135*$^{FUN025/FUN025}$ and Tg-Tmem135 MFs. One possible mechanism is that TMEM135 may be involved in inhibition of mitochondrial fusion. We tested this hypothesis by promoting mitochondrial fusion through overexpression of mitochondrial fusion factor, mitofusin 2 (*MFN2*) in WT and Tg-Tmem135 MFs. If TMEM135 inhibits mitochondrial fusion, we would see less elongated mitochondria in Tg-Tmem135 cells compared to WT cells upon MFN2 overexpression. However, we observed similar number of elongated (with lower MFN2 expression) and aggregated (with higher MFN2 expression) mitochondria in WT and Tg-Tmem135 MFs transfected with p*MFN2-YFP* (*Figure 5—figure supplement 2A*). The results indicate that overexpression of *Tmem135* in Tg-Tmem135 MFs does not inhibit the mitochondrial morphological changes caused by overexpression of *Mfn2*, suggesting that TMEM135 does not inhibit fusion. While it is also possible that downregulation of mitochondrial fusion proteins is responsible for over-fragmented mitochondria in Tg-TMEM135 MFs, which can be overcome by MFN2 overexpression, we found that this is not the case. The protein levels of mitochondrial fusion proteins, optic atrophy 1 (OPA1), MFN1 and MFN2 were not changed in Tg-TMEM135 MFs compared with WT MFs (*Figure 5—figure supplement 2B–E*). Additionally, the levels of these proteins are either unchanged (OPA1 and MFN2) or decreased (MFN1) in *FUN025* MFs compared with WT MFs (*Figure 5—figure supplement 2B–E*), indicating that the overly fused mitochondrial network observed in *FUN025* MFs was not caused by up-regulation of these mitochondrial fusion proteins.

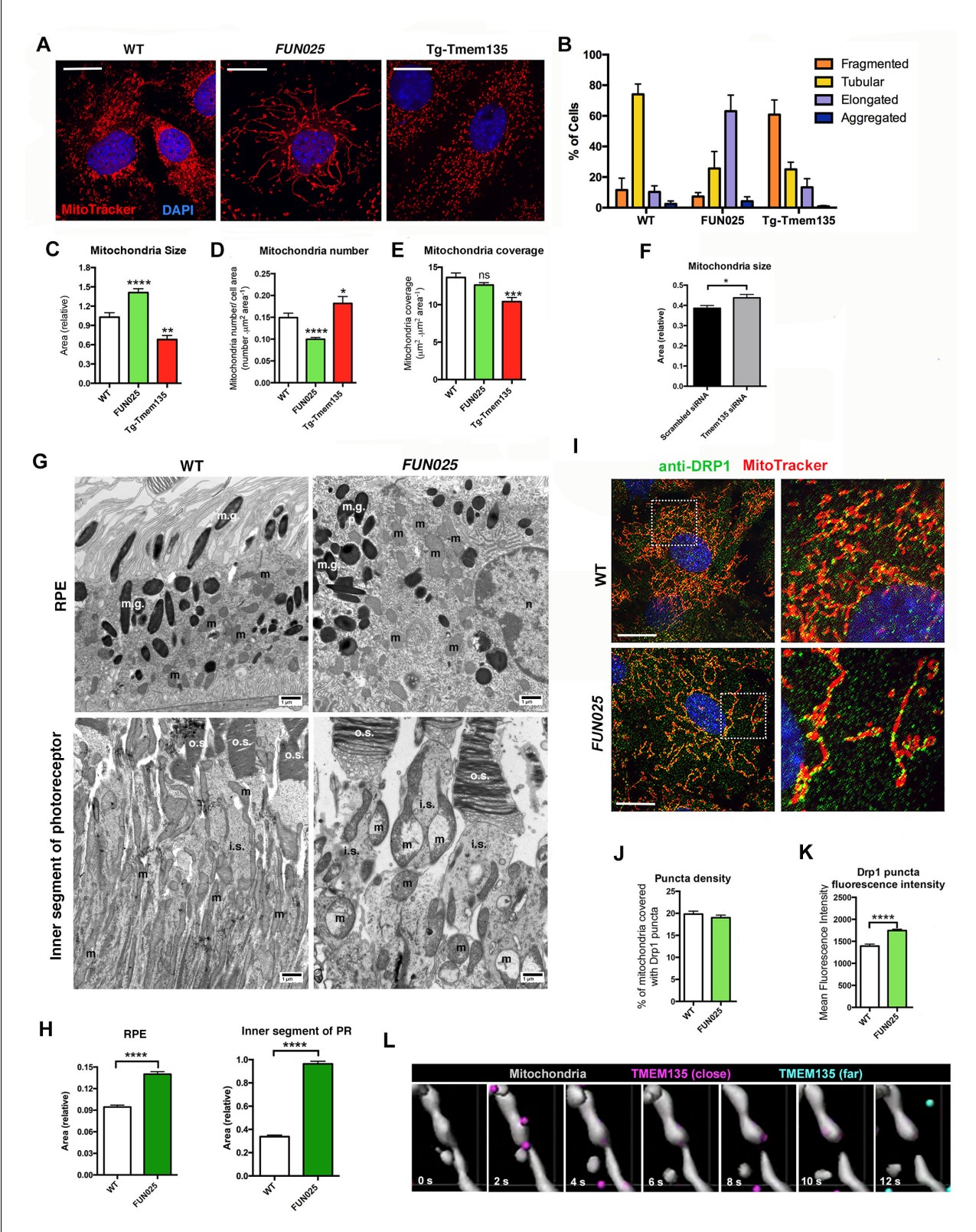

**Figure 5.** TMEM135 is involved in the balance of mitochondrial fission and fusion. (**A**) Morphology of mitochondria (MitoTracker; red) in WT, *Tmem135^FUN025/FUN025* (*FUN025*) and Tg-Tmem135 MFs. Scale bar = 10 μm. (**B**) Scoring of mitochondrial network morphologies in WT, *FUN025* and Tg-Tmem135 MFs. Data from *n* = 290 WT, *n* = 304 *FUN025*, and *n* = 372 Tg-Tmem135 cells; 3 mice per genotype. (**C–E**) Quantification of size, number and coverage of mitochondria in WT, *FUN025* and Tg-TMEM135 MFs. Data from *n* = 49 WT, *n* = 104 *FUN025*, and *n* = 29 Tg-Tmem135 cells; 3 mice per

*Figure 5 continued on next page*

*Figure 5 continued*

genotype. (F) Knocking down *Tmem135* by siRNA against *Tmem135* results in increased mitochondrial size in WT MFs. Data from $n = 72$ cells with scrambled siRNA, $n = 81$ cells with Tmem135 siRNA. Two tailed, unpaired, Student's *t*-test. (G) EM revealed the morphology of mitochondria in RPE and inner segments of photoreceptor cells from wild-type and *FUN025* mice. m = mitochondria; m.g. = melanin granules; n = necleus; o.s. = outer segments; i.s. = inner segments. Scale bar = 1 μm. (H) Quantification of mitochondria size in RPE and inner segments of photoreceptor cells from wild-type and *FUN025* mice. RPE data from 300 mitochondria from three wild-type mice and 400 mitochondria from four *FUN025* mice. Inner segments data from 200 mitochondria from four wild-type mice and 200 mitochondria from four *FUN025* mice. (I) Immunofluorescence of DRP1 (anti DRP1 antibody, green) and mitochondria (MitoTracker, red) in WT and *FUN025* MFs. Scale bar = 10 μm. (J–K) Quantification of DRP1 puncta density and fluorescence intensity on mitochondria. (L) Time-lapse fluorescence imaging (modified as described in Materials and methods) of a living HT22 cell expressing TMEM135-GFP and labeled with MitoTracker at the indicated time points. TMEM135 puncta that are located relatively close to the mitochondrial surface are shown in magenta, whereas those relatively far from the mitochondrial surface are shown in turquoise. Mitochondria (gray) was made partial transparent in order to see TMEM135 puncta located on the back of mitochondria. *p<0.05, **p<0.01, ***p<0.001, ****p<0.0001, two-way ANOVA. All data are mean ± s.e.m.

The following figure supplements are available for figure 5:

**Figure supplement 1.** Generation of Tg-TMEM135 mice.

**Figure supplement 2.** TMEM135 does not inhibit mitochondrial fusion.

**Figure supplement 3.** A portion of TMEM135 is colocalized with DRP1 on the mitochondria.

Another possible mechanism is that TMEM135 may promote mitochondrial fission. Dynamin related protein 1 (DRP1; also known as dynamin 1-like or DNM1L) is a mitochondrial dynamin-like GTPase that is essential for mitochondrial fission (*Mears et al., 2011*). During mitochondrial fission, DRP1 is recruited from the cytosol onto the mitochondrial outer membrane, where it is assembled into puncta (*Mears et al., 2011*). These puncta consist of oligomeric DRP1 complexes that wrap around and constrict the mitochondrial tubule to mediate fission (*Mears et al., 2011*). We found that some of the punctate DRP1 signals colocalize with TMEM135 puncta on mitochondria (*Figure 5—figure supplement 3*), suggesting that TMEM135 may be involved in DRP1-dependent mitochondrial fission. We first tested if DRP1 binding to mitochondria is affected in the *Tmem135*[FUN025/FUN025] MFs using immunofluorescence to visualize DRP1 puncta on mitochondria. In MFs, much of the DRP1 staining was diffused in the cytosol, but a proportion could be found in punctate structures on mitochondria (*Figure 5I*). We used image analysis to quantify the density (number of puncta/mitochondrial area) of DRP1 puncta on mitochondria (*Figure 5J*). The density of DRP1 puncta detected on mitochondria in *Tmem135*[FUN025/FUN025] MFs was not different from what was detected in WT MFs (*Figure 5J*), indicating that DRP1 translocation to the mitochondria was not affected in *Tmem135*[FUN025/FUN025] MFs compared to WT MFs. The average total fluorescence per puncta (*Figure 5K*) was even increased in *Tmem135*[FUN025/FUN025] MFs while the mitochondria stayed fused. These results indicate that DRP1 is translocated to mitochondria normally and assembled into higher order structures but appears to stay inactive in *Tmem135*[FUN025/FUN025] MFs. DRP1 has been shown to be activated or inactivated through various post-translational modifications at different amino acids (*Knott et al., 2008*). These results support the notion that TMEM135 may be required for DRP1 activation. To further characterize the role of TMEM135 in morphological

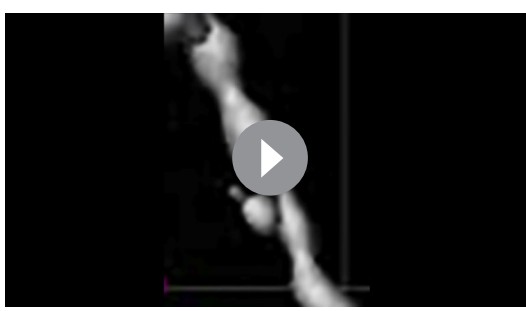

**Video 1.** Live imaging of HT22 cells expressing TMEM135-GFP. Live imaging of a HT22 cell expressing TMEM135-GFP and labeled with MitoTracker Red. TMEM135 puncta that are located relatively close to the mitochondrial surface are shown in magenta, whereas those relatively far from the mitochondrial surface are shown in turquoise. Mitochondria (gray) was made partial transparent in order to see TMEM135 puncta located on the back of mitochondria.

changes of mitochondria, we monitored their dynamics in relation to mitochondria in living WT MFs and HT22 cells expressing GFP-TMEM135. We observed that a proportion of TMEM135 localized at mitochondrial constriction sites where mitochondria later divided (*Figure 5L*; *Video 1*; *Video 2*). Based on these data, we conclude that TMEM135 is more likely involved in mitochondrial fission rather than fusion.

## TMEM135 plays a role in mitochondrial metabolism

We then examined if the structural changes of mitochondria in *Tmem135*<sup>FUN025/FUN025</sup> and Tg-Tmem135 mice affect their mitochondrial functions. First, we analyzed the mitochondrial membrane potential ($\Delta\Psi$M), and found that Tg-Tmem135 MFs showed a significantly increased $\Delta\Psi$M and *Tmem135*<sup>FUN025/FUN025</sup> MFs retained a similar $\Delta\Psi$M as WT MFs (*Figure 6A*; *Figure 6—figure supplement 1*). These results indicate that *Tmem135*<sup>FUN025/FUN025</sup> and Tg-Tmem135 MFs do not lose the $\Delta\Psi$M, suggesting that the individual unit of the mitochondrial membrane is functional. Next we determined whether impaired mitochondrial dynamics lead to total mitochondrial respiratory impairment within the cells (*Figure 6B*). Compared to WT MFs, *Tmem135*<sup>FUN025/FUN025</sup> and Tg-Tmem135 MFs showed a lower basal oxygen consumption rate (OCR) (*Figure 6C*). A significantly lower ATP production was detected in *Tmem135*<sup>FUN025/FUN025</sup> and Tg-Tmem135 MFs (*Figure 6D*). *Tmem135*<sup>FUN025/FUN025</sup> MFs showed a decrease in the maximal respiration and spare respiratory capacity (SRC), whereas Tg-Tmem135 showed no difference in these parameters (*Figure 6E,F*). The proton leak was not changed in *Tmem135*<sup>FUN025/FUN025</sup> and Tg-Tmem135 MFs (*Figure 6G*). Our results suggest that the differences in mitochondrial basal and maximal respiration could be due to the change of mitochondrial dynamics. SRC is a measure of the extra capacity available in cells to produce energy in response to increased stress or work (*van der Windt et al., 2012*). Our observation that *Tmem135*<sup>FUN025/FUN025</sup> MFs have significantly reduced SRC suggests that the mutant TMEM135 not only affects oxidative phosphorylation at the normal condition but also decreases the potential ability for the mitochondria to produce more energy under stressed conditions.

## Mutation and overexpression of TMEM135 both lead to increased oxidative stress

Dysfunctional mitochondria can generate more ROS, if not being eliminated instantly, leading to accelerated oxidative damage. We measured both total ROS and superoxide, which is the principal ROS formed by the mitochondrial electron transport chain (ETC), in the MFs. Both *Tmem135*<sup>FUN025/FUN025</sup> and Tg-Tmem135 MFs showed higher signals for total ROS and superoxide compared to WT MFs (*Figure 6H,I*; *Figure 6—figure supplement 2*). We then compared the protein level of major antioxidant enzymes, including superoxide dismutases (SODs), glutathione peroxidases 1 (GPx1) and catalase (CAT), in WT and *Tmem135*<sup>FUN025/FUN025</sup> MFs under normal and stress conditions (by adding hydrogen peroxide at 200 µM for 2 hr). The stress condition was created by exogenous exposure to hydrogen peroxide, which has been reported to activate the cells to produce superoxide via NAD(P)H oxidase (*van Klaveren et al., 1997*). The western blot results indicate that under the normal condition, the protein levels of SOD1, SOD2, SOD3, GPx1 and CAT are all increased in *Tmem135*<sup>FUN025/FUN025</sup> MFs compared to WT MFs (*Figure 6J*) reflecting an increase of ROS in *Tmem135*<sup>FUN025/FUN025</sup> MFs. As a positive control, WT MFs treated with hydrogen peroxide showed an increase in SODs and GPx1 but not CAT (*Figure 6J*). In contrast, none of the tested antioxidant enzymes except GPx1 showed an increase in hydrogen peroxide-treated *Tmem135*<sup>FUN025/FUN025</sup> MFs compared to non-treated *Tmem135*<sup>FUN025/FUN025</sup> MFs (*Figure 6J*). These results suggest that the

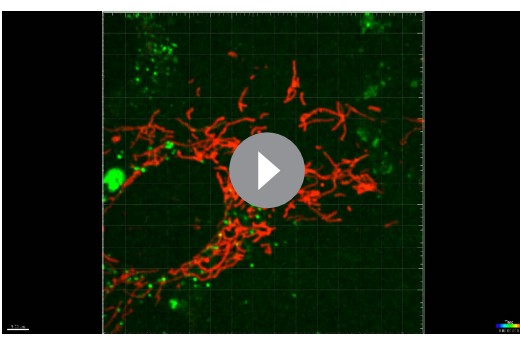

**Video 2.** Live imaging of WT fibroblasts expressing TMEM135-eGFP. Live imaging of fibroblasts derived from B6 (WT) mice transfected with TMEM135-eGFP plasmid and labeled with MitoTracker Red. TMEM135-eGFP are shown in green and mitochondria are shown in red.

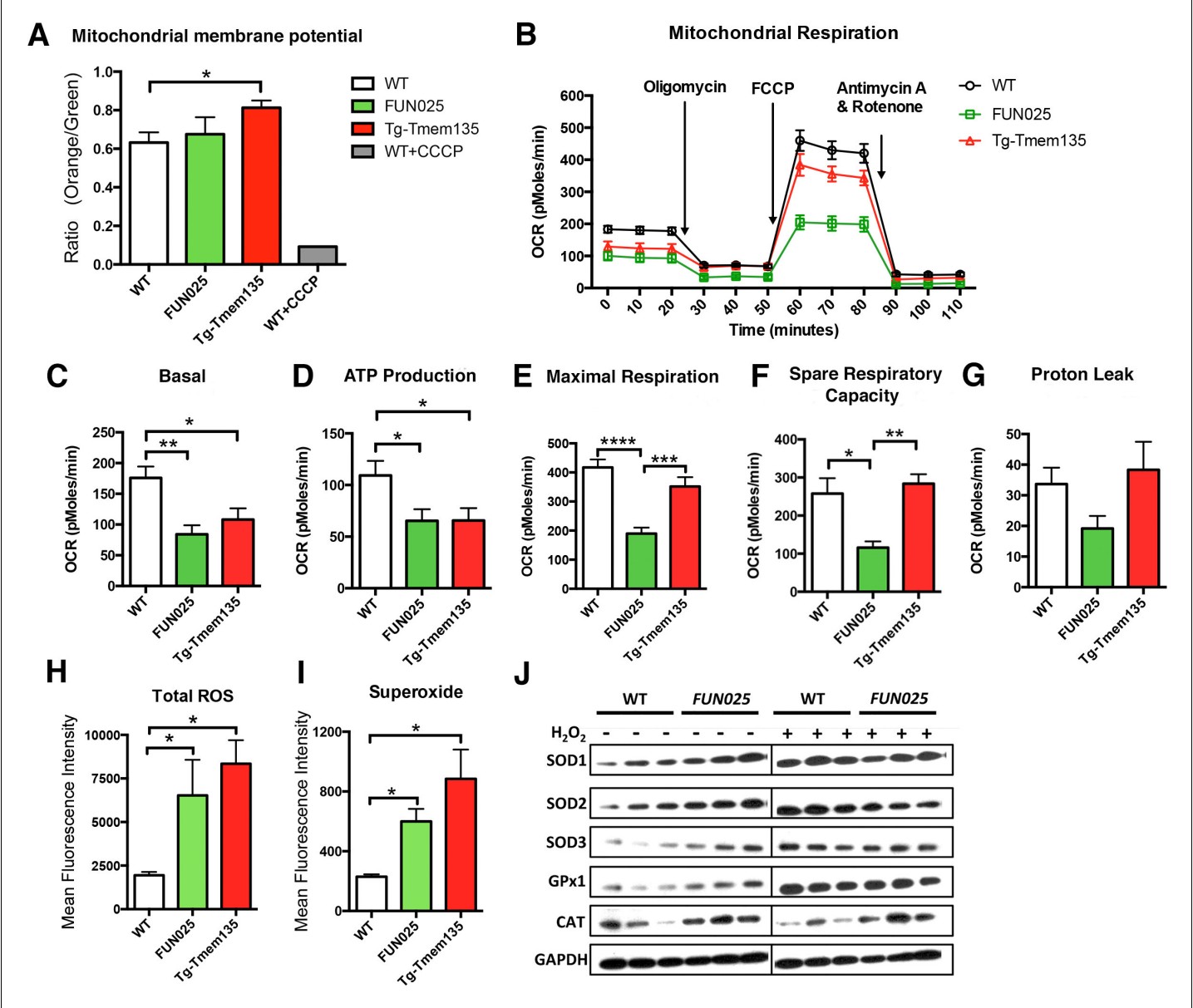

**Figure 6.** *Tmem135* plays a role in mitochondrial metabolism. (**A**) Mitochondrial membrane potential (ΔΨM) in Tg-Tmem135 MFs was higher than that in WT MFs, whereas ΔΨM in *Tmem135^FUN025/FUN025^* (*FUN025*) MFs was comparable to that in WT MFs. Data from *n* = 3 mice per genotype. *n* = 1 for WT+CCCP. (**B–G**) Oxygen consumption rates (OCR) of MFs were determined with a Seahorse XF^e^24 Extracellular Flux Analyzer in basal and stimulated conditions (n = 3 mice per genotypes). The areas under the curve from different sections of the experiment (**B**) are shown as individual histograms for basal respiration, ATP production, maximal respiration, spare respiratory capacity, and proton leak (**C–G**). Data from *n* = 3 mice per genotype. (**H–I**) Total ROS and superoxide in *FUN025* and Tg-Tmem135 MFs were higher compared to WT cells. Data from *n* = 3 mice per genotype. (**J**) Western blot analysis for SOD1, SOD2, SOD3, GPx1and CAT in WT and *FUN025* MFs with and without the hydrogen peroxide treatment. Protein levels were normalized using GAPDH expression levels. *p<0.05, **p<0.01, ***p<0.001, ****p<0.0001, two-way ANOVA. All data are mean ± s.e.m..

The following figure supplements are available for figure 6:

**Figure supplement 1.** Mitochondrial membrane potential in *FUN025* and Tg-Tmem135 MFs.

**Figure supplement 2.** Increased ROS and superoxide in *FUN025* and Tg-Tmem135 MFs.

antioxidant system is upregulated in response to the increased ROS in Tmem135$^{FUN025/FUN025}$ MFs. However, they may have reached the maximal capacity to upregulate the antioxidant system, and unable to further upregulate it upon exposure to exogenous oxidative stress.

## Tmem135$^{FUN025/FUN025}$ mice are more sensitive to the hyperoxic condition

Our cellular data showed that Tmem135$^{FUN025/FUN025}$ MFs had increased ROS, decreased SRC and the antioxidant system incapable of responding to additional environmental ROS. For that reason, we hypothesized that Tmem135$^{FUN025/FUN025}$ mice may be more sensitive to oxidative stress in vivo due to the higher basal ROS level and reduced capacity to produce more energy under a stressed condition. Hyperoxia has been used as a condition to induce oxidative stress in different organs (*Jamieson et al., 1986*). We exposed two-month-old Tmem135$^{FUN025/FUN025}$ and control mice to the hyperoxic condition (75% O$_2$, 14 days) to test the effect of oxidative stress to the mutant phenotypes. TMEM135$^{FUN025/FUN025}$ retina were more susceptible to oxidative stress-induced cell death than WT mice as indicated by the ONLT index and the number of terminal dUTP nick end labeling (TUNEL) positive cells (*Figure 7A,B*). While retinal stress marker, GFAP increased in both control and Tmem135$^{FUN025/FUN025}$ retina under the hyperoxic condition (*Figure 7C*), Tmem135$^{FUN025/FUN025}$ mice under the hyperoxic condition had the most severe phenotype (*Figure 7C*). In both WT and Tmem135$^{FUN025/FUN025}$ retina, hyperoxia resulted in upregulation of 4-hydroxy-2-nonenal (4-HNE) (*Figure 7D*), a commonly used marker of lipid peroxidation (*Cingolani et al., 2006*). In addition, the TMEM135 protein level increased in the Tmem135$^{FUN025/FUN025}$ retina compared to WT control in the normal air with normal inspired PO2 (*Figure 7D*). In Tmem135$^{FUN025/FUN025}$ but not WT retina, hyperoxia resulted in an upregulation of the TMEM135 protein level (*Figure 7D*). Together, these data suggest that TMEM135 plays a role in protecting cells from increased oxidative stress.

## Discussion

In this study, we attempt to address a central question regarding age-dependent diseases: 'Why do age-dependent diseases manifest themselves in an age-dependent manner?' While it can be postulated that disease-causing mechanisms interact with molecular/cellular changes that occur with aging, molecules and pathways linking the aging process and age-dependent diseases have not been identified. By using a forward genetics approach, we identified a novel mouse mutation that leads to accelerated aging as well as pathologies observed in age-dependent retinal disease (*Figure 1*; *Figure 2*). Since this 'phenotype-driven' approach is unbiased and requires no prior knowledge of the gene functions, it allows the discovery of unsuspected mechanisms underlying biological phenomena and diseases. The mutation was discovered in transmembrane protein 135 (TMEM135), which has been associated with fat storage and longevity in *C. elegans* but has no defined molecular functions. Our results suggest that TMEM135 is critical for the regulation of the onset and progression of the aging process, a defect in which leads to age-dependent diseases. Thus, TMEM135 presents a novel molecular link between the aging process and age-dependent diseases.

As stated above, the unique characteristics of Tmem135$^{FUN025/FUN025}$ mice is that it displays phenotypic features associated with normal retinal aging at much younger ages as well as age-dependent disease pathologies. The normal aging phenotypes in the mouse retina includes photoreceptor cell degeneration, ectopic synapse formation, and increased retinal stress (*Higuchi et al., 2015*). While these age-dependent abnormalities are observed at 8 months on an aging-susceptible A/J background and later in a less susceptible B6 background (*Higuchi et al., 2015*), Tmem135$^{FUN025/FUN025}$ mice on the B6 background have an early onset (as early as 2 months of age) and more rapid advancement of the retinal aging phenotypes (*Figure 1*). We previously found that these retinal aging phenotypes all start from the peripheral retina and progress toward the central retina (*Higuchi et al., 2015*). This spatial pattern of progression is unique to the retinal aging phenotypes in mice, and is maintained in the Tmem135$^{FUN025/FUN025}$ retina albeit with earlier onset and faster progression, indicating that the retinal aging process is accelerated in Tmem135$^{FUN025/FUN025}$ mice. On the other hand, these mice also display pathologies observed in an age-dependent retinal disease, AMD, including autofluorescent aggregates, increased inflammation and thickened RPE (*Figure 2*). The fact that a single gene mutation leads to both accelerated aging and age-dependent

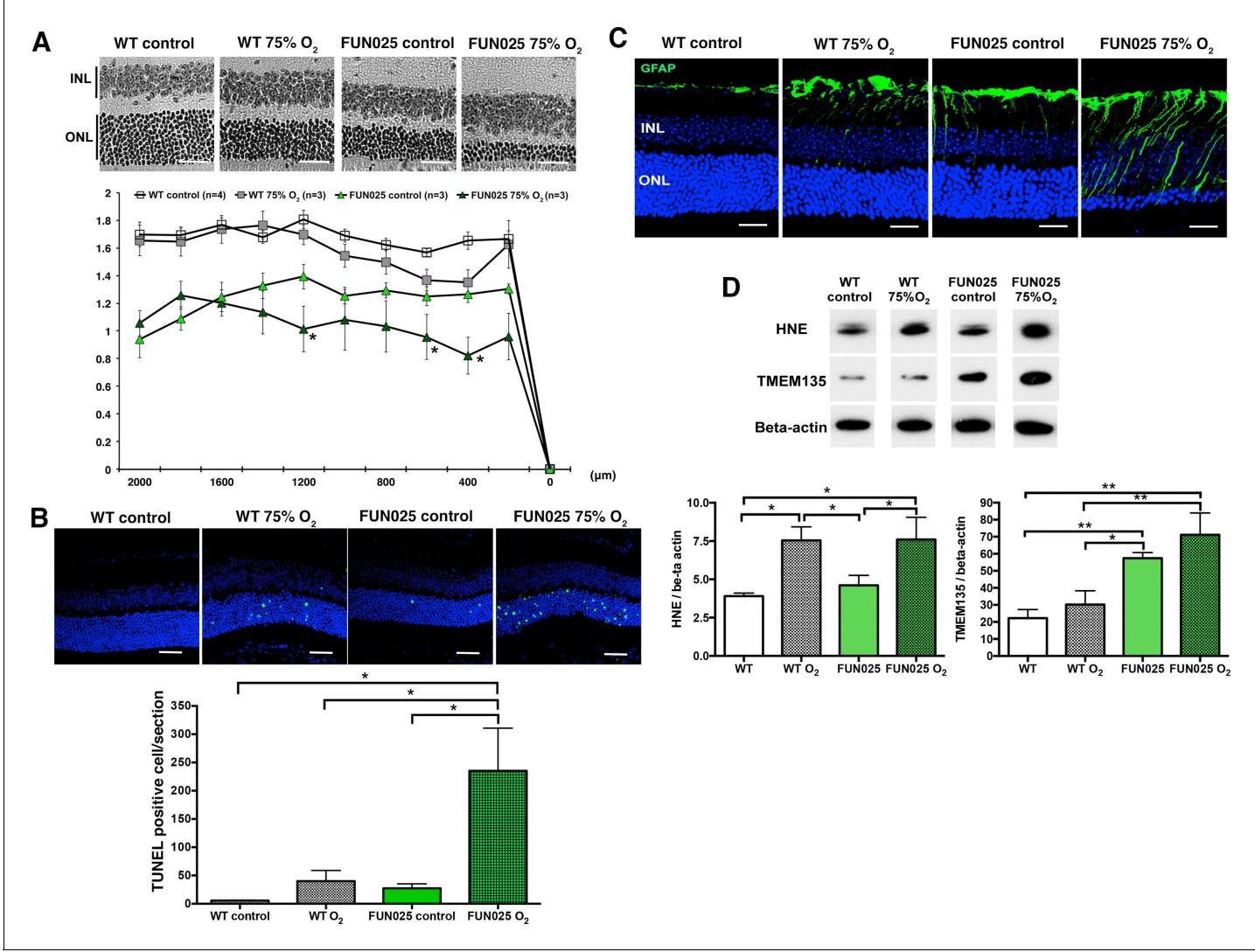

**Figure 7.** *Tmem135[FUN025/FUN025]* mice are more sensitive to hyperoxic condition. (A–B) *Tmem135[FUN025/FUN025]* (FUN025) mice raised in 75% $O_2$ for two weeks show significant decrease of ONLT and increase of TUNEL positive cells, indicating accelerated photoreceptor cell death by apoptosis. Data from $n$ = 4 WT mice in control air, $n$ = 3 WT mice in 75% $O_2$, $n$ = 3 *FUN025* mice in control air, and $n$ = 3 *FUN025* mice in 75% $O_2$. Scale bar = 20 µm. (C) Upregulation of GFAP (green) indicating retinal stress is observed in *FUN025* mice raised in the normal air, as well as WT mice and *FUN025* mice raised in 75% $O_2$ for two weeks. *FUN025* mice raised in 75% $O_2$ have the highest increase of GFAP signals in the retina. Scale bar = 20 µm. (D) Western blotting showing that hyperoxia results in upregulation of 4-HNE in both WT and *FUN025* retina, and an increase of TMEM135 in *FUN025* retina but not in WT retina. Data from $n$ = 3 WT mice in control air, $n$ = 3 WT mice in 75% $O_2$, $n$ = 3 *FUN025* mice in control air, and $n$ = 3 *FUN025* mice in 75% $O_2$. *$p<0.05$, **$p<0.01$, two-way ANOVA. All data are mean ± s.e.m.

disease pathologies in this mouse model provides a confirmation that these processes are closely associated with each other at the molecular level, and supports the idea that accelerating the aging process may lead to age-dependent diseases. Consistent with this idea, pathologies similar to those in the aging human retina including drusen formation, ectopic synapse formation and photoreceptor cell degeneration are observed in AMD patients with earlier onset and/or increased severity (*Eliasieh et al., 2007*; *Hageman et al., 2001*; *Liets et al., 2006*; *Sullivan et al., 2007*; *Terzibasi et al., 2009*). Based on our finding, TMEM135 may be a key molecule that could tip the normal aging process toward age-dependent diseases.

Our observations associated with the RPE in *Tmem135[FUN025/FUN025]* mice indicate that this cell layer may be the primary site affected by the *Tmem135[FUN025]* mutation. We observed subretinal autofluorescent cells in the *Tmem135[FUN025/FUN025]* retina that are positive for microglia and

macrophage markers. Microglia have been reported to sense and react to different stimuli, and move toward the origin of stress or the injury site (*Mcglade-Mcculloh et al., 1989*) while going through multiple morphological stages of microglial activation (*Jonas et al., 2012*). Upon evaluating the morphology and orientation of microglia in the *Tmem135$^{FUN025/FUN025}$* retina, we found that the microglia located closer to the RPE (and thus farther from the IPL) showed increased body size and decreased cell processes compared to those located closer to the IPL (*Figure 2C*). Similar morphological changes of microglia were documented in the injured rat olfactory bulb, and were classified as stages of microglial activation response to stimuli (*Jonas et al., 2012*). The Iba1$^+$ cells near the apical surface of the RPE also show immunoreactivity to F4/80, indicating that they have transformed from microglia to macrophages (*Figure 2C*). These data suggest that the RPE layer is the origin of stress, toward which the microglia migrate. In addition, in the *Tmem135* mutant retina, RPE abnormalities were observed earlier than the onset of photoreceptor cell degeneration. Since the RPE is located between the choriocapillaris/Bruch's membrane and the photoreceptors, and maintains the health of photoreceptor cells by delivering oxygen and metabolites and phagocytizing the shed photoreceptor outer segment discs (*Sparrow et al., 2010*), primary RPE defects could result in secondary loss of photoreceptor. Interestingly, this is consistent with the current hypothesis for pathogenesis of AMD (*Bonilha, 2008*; *Roth et al., 2004*). Together, our data suggest that the RPE and the subretinal space are likely affected first and become the origin of stress followed by photoreceptor cell loss in *Tmem135$^{FUN025/FUN025}$* mice.

In the cell culture study, we revealed that the *Tmem135$^{FUN025/FUN025}$* mutant cells exhibit overfused mitochondrial networks whereas Tg-Tmem135 cells (transgenic cells overexpressing WT *Tmem135*) show over-fragmented mitochondrial networks. Since mitochondrial networks are the net product of fusion and fission events that occur within cells, there are two possible mechanisms TMEM135 can be involved in: inhibition of mitochondrial fusion or promotion of mitochondrial fission. Following observations in our study strongly indicated that TMEM135 in involved in promotion of mitochondrial fission rather than inhibition of mitochondrial fusion. First, experimentally promoting mitochondrial fusion through overexpression of mitochondrial fusion factor MFN2 in WT and Tg-Tmem135 cells resulted in similarly over-fused mitochondrial networks in WT and Tg-Tmem135 cells (*Figure 5I*) indicating that TMEM135 does not inhibit mitochondrial fusion. Second, we observed that some of the TMEM135 signals are colocalized with punctate signals of the mitochondrial fission factor, DRP1, on mitochondria (*Figure 5—figure supplement 3*). Additionally, we found that DRP1 is normally translocated onto mitochondria which stay fused in *Tmem135$^{FUN025/FUN025}$* mutant cells, indicating that DRP1 is not properly activated to promote fission. It has been shown that DRP1 activity is regulated delicately by a number of post-translational modifications including phosphorylation, *S*-nitrosylation, SUMOylation, ubiquitination, and O-GlcNAcylation (*Cereghetti et al., 2008*; *Chang and Blackstone, 2010*; *Knott et al., 2008*; *Zunino et al., 2007*). Hence, our results suggest an important role for TMEM135 to promote DRP1-dependent fission possibly by activating DRP1 through post-transcriptional modification.

Another possible cause for over-fragmented mitochondrial networks observed in Tg-Tmem135 cells is increased oxidative stress, which has been shown to induce mitochondrial fragmentation in cultured cells (*Iqbal and Hood, 2014*; *Wu et al., 2011*). Although we cannot completely rule out this possibility, it is unlikely that oxidative stress is the major cause of mitochondrial fragmentation in Tg-TMEM135 cells. While ROS is increased in both *Tmem135$^{FUN025/FUN025}$* mutant cells and Tg-TMEM135 cells (*Figure 6H,I*), mitochondrial networks are over-fused in mutant cells and over-fragmented in Tg-TMEM135 cells (*Figure 5A–E*), suggesting the TMEM135 function rather than oxidative stress as the primary factor affecting the mitochondrial networks in these cells. In addition, our live imaging of GFP-TMEM135 in WT MFs and HT22 cells captured the direct interaction of TMEM135 with mitochondria at fission sites, providing additional evidence that the TMEM135 function in mitochondrial fission rather than oxidative stress is the cause of altered mitochondrial networks.

We report here that defects in a novel regulator of mitochondrial dynamics, TMEM135, lead to impaired mitochondrial respiration. We found that adult mouse fibroblasts with either over-fused (*Tmem135$^{FUN025/FUN025}$* mutant) or over-fragmented (Tg-TMEM135) mitochondria show impairment of respiration, indicating that the balance between mitochondrial fission and fusion is important for proper respiratory functions. In addition to lower basal respiration and ATP production, *Tmem135$^{FUN025/FUN025}$* mutant fibroblasts also showed decreased SRC. SRC or 'spare respiratory capacity'

refers to the difference between basal and maximal respiration, which reflects the extra mitochondrial capacity available in a cell to produce energy under conditions of increased work or stress (*Nicholls, 2009*). Exhaustion of the SRC has been associated with heart diseases, neurodegenerative disorders and smooth muscle death (*Hill et al., 2010*; *Nicholls, 2008*; *Sansbury et al., 2011*; *Yadava and Nicholls, 2007*). It has been also hypothesized that mitochondria contributes to aging and age-dependent pathologies through a life-long continued decrease of the SRC (*Desler et al., 2012*). The $Tmem135^{FUN025/FUN025}$ mutation that leads to decreased SRC (*Figure 6F*) and accelerated retinal aging phenotypes provides new evidence to support this hypothesis. In addition, both $Tmem135^{FUN025/FUN025}$ and Tg-Tmem135 MFs have increased total ROS and superoxide compared to WT MFs (*Figure 6H,I*; *Figure 6—figure supplement 2*), indicating increased oxidative stress and damage. Based on the decreased SRC and increased ROS in $Tmem135^{FUN025/FUN025}$ MFs, we hypothesized that the $Tmem135^{FUN025/FUN025}$ retina may be more vulnerable to the increase of environmental stress. Our animal study showed that $Tmem135^{FUN025/FUN025}$ mice under the hyperoxic condition had more severe retinal degeneration, apoptosis in the retina and retinal stress compared to WT mice under the hyperoxic condition (*Figure 7A–C*). These results suggest that while WT mice can tolerate certain amount of oxidative stress, $Tmem135^{FUN025/FUN025}$ mutant mice with a decline in mitochondrial functions have lower tolerance to such stress. Thus, $Tmem135^{FUN025/FUN025}$ mutant mice are more susceptible to damages caused by environmental stressors, accumulation of which leads to accelerated aging and age-dependent diseases with early onset. In conclusion, both our cell and animal studies suggest a role for TMEM135 in protecting cells from increased environmental stress.

In summary, through genetic analysis of an ENU-induced mutant mouse strain, we have identified that TMEM135 plays a critical role in maintaining the health of the retina by keeping the balance of mitochondrial dynamics, defects in which result in accelerated retinal aging and development of age-dependent disease pathologies. In particular, our data suggest that TMEM135 may promote DRP1-dependent fission through activation of DRP1. Our study suggests that TMEM135 or consequences of its defect such as unbalanced mitochondria dynamics and increased oxidative stress could be potential therapeutic targets for retinal aging and age-dependent diseases.

## Materials and methods

### Animals

All experiments were performed in accordance with the National Institute of Health Guide for the Care and Use of Laboratory Animals and were approved by the Animal Care and Use Committee at the University of Wisconsin-Madison and Northwestern University. *FUN025* mice were generated by ENU mutagenesis in the Northwestern University Center for Functional Genomics as described previously (*Pinto et al., 2004*; *Vitaterna et al., 2006*) and isolated from a screen designed to detect recessive mutants with vision phenotypes. These mice on the C57BL/6J genetic background were imported to University of Wisconsin-Madison. Affected *FUN025* mice were crossed to C57BL/6J wild-type mice and the offspring were crossed to affected mice. Subsequently, the mutant line was maintained by crossing affected (based on phenotyping by fundus photography) mice to non-affected siblings, which were presumably heterozygous for the recessive mutation. C57BL/6J (RRID: IMSR_JAX:000664) mice obtained from The Jackson Laboratory were used as control mice in the experiments. Tg-Tmem135 mice were generated at University of Wisconsin-Madison. We replaced the EGFP sequence in the pCX-EGFP vector (kindly provided by Dr. Junichi Miyazaki) (*Niwa et al., 1991*) with the full length *Tmem135* cDNA and named it pCX-TMEM135. We used pCX-TMEM135 for the transgene construct after linearization with HindIII and SalI (New England Biolabs, Ipswich, MA). The construct was micro-injected into pronuclei of FVB/NJ embryos at the Transgenic Facility of the University of Wisconsin-Madison Biotechnology Center. Transgene-positive founders were crossed to C57BL/6J mice for two generations and subsequently maintained by intercrossing. The $Ped6b^{rd1}$ mutation in the FVB/NJ background was removed during this process. Mice with a point mutation (T > C) in *Tmem135* that is the same as observed in *FUN025* mice were generated using the CRISPR/Cas9 system (T > C mice). The T > C mutation was introduced into intron 12 of *Tmem135* (Chr7:96,296,478) in C57BL/6J mice at Translational Genomics Facility of University of Wisconsin-Madison Biotechnology Center. 'Optimized CRISPR Design' (http://crispr.mit.edu/) was

used to choose the sequence (5'-GCCAAGCACACAGGGTTTGC-3') that is located at 28 to 47 nucleotides downstream of the mutation site. This DNA sequence was incorporated into the px330 plasmid (*Cong et al., 2013*) (kindly provided by Feng Zhang) for generation of the single guide RNA (sgRNA). In vitro transcription template was generated by PCR using the plasmid and an oligo with a T7 adapter end, and the sgRNA was transcribed using the MEGAshortscript T7 Transcription Kit (Thermo Fisher Scientific, Waltham, MA) and purified with the MEGAclear Kit (Thermo Fisher Scientific, Waltham, MA). In addition, we used a 200 nucleotide oligo DNA containing the point mutation (T > C) in the middle of the sequence as a donor oligo (5'-AGTTTTTCCTTGTCATTTCAGGG TTTTTGGCAGGTGTGTCGATGATGTTTTATAAAAGCACAACAATTTCCATGTACCTAGCTTCCAAGC TGGTGGAGGcAAGCACAGCTCTTATGCCTGAGAAGTTGCCAAGCACACAGGGTTTGCAGGTGG TGTGGAGTTGCTTCAGTTGCAAGAATGACTGCTACCAAAGCAGCTCT -3'). The Cas9 protein (40 ng/µl), sgRNA (50 ng/µl) and donor oligo (50 ng/µl) were injected into C57BL/6J embryos at Transgenic Facility of University of Wisconsin-Madison Biotechnology Center. Following microinjection, the embryos were transferred into the oviducts of pseudo-pregnant recipients. We obtained 4 mutants with the point mutation.

## Outer nuclear layer thickness (ONLT) measurement

The thickness of retinal layers was measured in H&E stained sections by using the Measure function of ImageJ software (available at http://rsb.info.nih.gov/ij; developed by Wayne Rasband, National Institutes of Health). ONLT index was calculated as the ONL thickness normalized to the INL thickness.

## Histological analysis

Following asphyxiation of mice by $CO_2$ administration, eyes were immediately removed and immersion fixed in Bouin's fixative overnight at 4°C. Eyes were then rinsed, dehydrated, and embedded in paraffin. Paraffin blocks were sectioned 6 µm thick on an RM 2135 microtome (Leica Microsystems, Wetzlar, Germany) and mounted on glass slides. The slides were then stained with hematoxylin and eosin (H&E) to visualize the retinal structure. H&E-stained sections were imaged on an Eclipse E600 microscope (Nikon, Tokyo, Japan, using a SPOT camera (Spot Diagnostics, Sterling Heights, MI).

## Immunohistochemistry

For cryostat sections, eyes were fixed in 4% paraformaldehyde (PFA) for 2 hr at 4°C, then cryoprotected at 4°C in a graded series of sucrose. Eyes were embedded in optimal cutting temperature compound (OCT) (Sakura Finetek USA, Torrance, CA) and sectioned at 12 µm thickness. For immunohistochemistry on cryostat sections, sections were blocked in PBS with 0.5% Triton X-100 and 2% normal donkey serum for 1 hr at room temperature. Next, sections were incubated overnight with the primary antibody against TMEM135 (Sigma-Aldrich Cat# SAB2102454, RRID:AB_10611002, St. Louis, MO. This antibody has been discontinued recently from Sigma but the same antibody is available from Aviva Systems Biology Cat# ARP49773_P050, RRID:AB_2048451, San Diego, CA), PKCα (Sigma-Aldrich Cat# P4334, RRID:AB_477345, St Louis, MO), PSD95 (UC Davis/NIH NeuroMab Facility Cat# 75–028, RRID:AB_2292909, Davis, CA), Iba1 (Wako Cat# 019–19741, RRID:AB_839504, Richmood, VA), F4/80 (Abcam Cat# ab6640, RRID:AB_1140040, Cambridge, MA), GFAP (Lab Vision Cat# RB-087-A0, RRID:AB_60417), CRALBP (Abcam Cat# ab15051, RRID:AB_2269474, Cambridge, MA), TOMM20 (Sigma-Aldrich Cat# WH0009804M1, RRID:AB_1843992, St. Louis, MO), NLRP3 (Abcam Cat# ab4207, RRID:AB_955792 Cambridge, MA), and caspase1 (Santa Cruz Biotechnology Cat# sc-514, RRID:AB_2068895, Dallas, TX). Sections were rinsed in PBS, and incubated with a 1:200 diluted Alexa 488 conjugated secondary antibody (Thermo Scientific, Rockford, IL) and/or Cy3 conjugated secondary antibody (Jackson ImmunoResearch Laboratories, West Grove, PA) for 45 min at room temperature. All sections were imaged on the Nikon A1R+ confocal microscope (Nikon Instruments, Melville, NY) equipped with high sensitivity GaAsP detectors; high-speed resonant scanner; six lasers at 405, 440, 488, 514, 561, and 640 nm. NIS-Elements AR software (Nikon Instruments, Melville, NY) was used for image acquisition and image analysis.

## Quantification of ectopic dendrites

Frequencies of ectopically localized bipolar cell dendrites extending into the ONL were quantified in sections immunostained with the PKCα antibody, using the Measure and Label function of ImageJ software. We counted the number of PKCα fiber that extended beyond the OPL and the length was measured along the outer plexiform layer (OPL), using the Measure function of ImageJ software. Frequency was calculated as the number of ectopic bipolar cell dendrites per millimeter of retina length.

## Fundus photography

Fundus photography was performed as previously described (*Pinto et al., 2004*). The iris was dilated with 1% Mydriacil and the corneas kept moist with saline solution during photography. The mouse was positioned on a heating pad during the procedure to maintain the body temperature. Photographs were made with a Kowa small animal fundus camera (2.5 megapixels) equipped with a 66 diopter supplemental lens (Kowa American Corporation, Torrance, CA).

## Electroretinogram (ERG)

ERG was performed on seven-month-old mice housed in standard diurnal cycling. Mice were dark adapted overnight and ERG was performed as previously detailed (*Pattnaik et al., 2015*). To determine cone response to light, we used a background light exposure of 30 cd.s/m2 during 10 min and stimulus flash intensities ranged from 0.01 to 25 cd.s/m2. To prevent cataracts, we used tear supplements during handling of mice. Animals were maintained at 37°C during the entire procedure. All ERG data was stored and exported in digital format for post-hoc analysis using Microsoft Excel.

## Genetic mapping

To map the *FUN025* gene, we performed a whole genome scan using F2 animals from mating (C57BL6-*FUN025* x C57BR/cdJ) F1 mice. We initially used 30 microsatellite markers, which distinguish C57BR/cdJ alleles from C57BL/6J alleles across the whole genome. All F2 animals were phenotyped by the presence or absence of drusen-like spots in the image produced by fundus photography, as previously described (*Pinto et al., 2007*). Once the chromosomal locus on chromosome 7 was identified for the *FUN025* mutation, we narrowed the genetic region by genotyping and phenotyping F2 recombinant mice from mating (C57BL7-*FUN025* x 129S1/SvImJ) F1 mice. We used microsatellite markers and SNPs to differentiate C57BL/6J and 129S1/SvImJ alleles. All marker positions reported are based on the NCBI mouse genome build 37.1 reference assembly.

## Genotyping

All genotyping was carried out by polymerase chain reaction (PCR). For *FUN025* genotyping, PCR primers, m*Tmem135* F1 (GGTTTTTGGCAGGTGTGTC) and m*Tmem135* R1 (TGTGTGCTTGGCAACTTCTC), were used for amplification of the wild-type (WT) allele and *FUN025* allele (118 bp). Cac8I (NewEngland Biolabs) was used to digest the *FUN025* allele specifically to generate two bands (80 bp and 38 bp).

## Targeted capture and next-generation sequencing

Using a SureSelect custom DNA bait library representing the *FUN025* region (Chr 7: 86,816,656 – 98,732,541) created by Agilent Technologies, sequence capture was performed on genomic DNA isolated from the spleen of *FUN025* mice (Qiagen DNEasy Blood and Tissue Kit, QIAGEN, Valencia, CA) followed by paired end sequencing on the Illumina HiSeq platform at DNA Sequencing Facility of University of Wisconsin-Madison Biotechnology Center. Alignment of the sequence reads to the C57BL/6J reference genome (NCBI build 37) was performed by the Bioinformatics Research Center at University of Wisconsin-Madison Biotechnology Center using Bowtie Software (http://bowtie-bio.sourceforge.net/index.shtml). The sequence reads cover 93% of bases within the candidate region, with an average coverage of greater than 100x. Highly repetitive regions were excluded from the capture array and are not represented here. Standard bioinformatic analysis of our sequencing data revealed 20 single nucleotide polymorphisms (SNPs) within five genes in the candidate region, all of which were intronic.

## Cell culture

All the primary cells and cell lines used in this study were subjected to morphology check by microscope and used at low passages in our laboratory. Authentication and quality control tests on distributed lots of cell lines were performed by the distributors. No cell lines used in this study are cross-contaminated or misidentified by International Cell Line Authentication Committee (ICLAC). Primary fibroblasts were harvested from three-month-old mouse ears. Briefly, two pieces of ear punched tissues were collected into a 1.5 ml microcentrifuge tube containing 70% ethanol, and the tissue pieces were rinsed with PBS containing penicillin and streptomycin. The tissues were diced into small pieces using a razor blade in a 6 cm Petri dish, and gathered into a microcentrifuge tube with 0.5 ml Trypsin-EDTA (0.25% Trypsin, 0.1% EDTA) (Thermo Fisher Scientific, Waltham, MA) and 0.5 ml Dispase (5 U/ml) (STEMCELL Technologies, Vancouver, Canada). The tissues were incubated at 37° C for 30 min, followed by centrifugation for 5 min at 3000 rpm. The supernatant was discarded and the tissues were washed with 2 ml HBSS. After centrifugation for 5 min at 3000 rpm, the supernatant was discarded. Then, 0.5 ml trypsin-EDTA was added to the precipitated cells. After mixing thoroughly, cells were incubated at 37° C for 20 min. Following incubation, the solution was centrifuged and the supernatant was decanted. The pellet was resuspended in 0.5 ml fibroblast culture media: Dulbecco's Modified Eagle's Medium (DMEM, ATCC, Manassas, VA) with 10% Fetal Bovine Serum (FBS, ATCC, Manassas, VA), 1% Penicillin Streptomycin (Thermo Fisher Scientific, Waltham, MA). Cell aggregates were triturated and cell suspension was plated into a 3 cm tissue culture dish with 2 ml fibroblasts culture medium. Cells were incubated at 37° C with 5% $CO_2$. Fibroblasts were sub-cultured every 2–4 days at 1:4 to 1:6 ratio. Cells were split similarly for expansion into T75 flasks for final harvest. Fibroblasts were tested for mycoplasma contamination and the results were negative. Primary hippocampal cell culture was performed using SPOT culture kit (University of Illinois at Chicago). COS-7 cells (ATCCCRL-1651, passage number 2–4; RRID:CVCL_0224) were obtained from a biological resource center, ATCC and used at low-passage. ATCC perform authentication and quality-control tests on all distribution lots of cell lines. COS-7 cells were maintained and sub-cultured according to the suggested culture methods from ATCC (www.atcc.org/). COS-7 cells were cultured in DMEM with 10% FBS. All of the medium and supplements mentioned above unless otherwise indicated were purchased from ATCC. Hippocampal cells were cultured in Neurobasal medium (Invitrogen, Thermo Fisher Scientific, Waltham, MA) with B27 supplement (Invitrogen, Thermo Fisher Scientific, Waltham, MA) and Glutamine (Invitrogen, Thermo Fisher Scientific, Waltham, MA) as the product manual recommendation. Immortalized mouse hippocampal cell line (HT22; RRID:CVCL_0321) was a gift from Dr. Kiren Rockenstein (Salk Institute, San Diego, CA). A stable cell line expressing TMEM135-GFP (pDEST53B6TMEM135) was established using HT22 cells. HT22-TMEM135-GFP stable cell line was maintained and sub-cultured in DMEM with 10% FBS. In the $H_2O_2$ study, fibroblasts were plated in 100 mm dishes at a concentration of ~1 × 10$^6$ cells per dish in the complete fibroblasts culture medium described above. Two days later, the cultured cells were exposed to 200 μM $H_2O_2$ (Sigma-Aldrich, St. Louis, MO) at 37°C for 2 hr and were immediately washed with cold PBS, lysed with cold RIPA buffer, protease inhibitor and phosphatase inhibitor (Thermo Scientific, Rockford, IL), centrifuged, and stored at −80°C until use.

## Transfection of mouse fibroblasts with expression vectors

Full length Tmem135 constructs were cloned into vector by pENTRTM Directional TOPO Cloning Kits (invitogen, Thermo Fisher Scientific, Waltham, MA) and were purified using a QIAprep Spin Miniprep Kit (QIAGEN, Valencia, CA) after culturing on the LB agar plate containing 10 mg/mL of kanamycin and in the LB liquid medium. pcDNADEST53 (GFP-attR1-CmR-ccdB-attR2; invitogen, Thermo Fisher Scientific, Waltham, MA) was used as the destination vector for TMEM135. The LR recombination reaction between the entry clone and a destination vector was carried out using LR Clonase Enzyme (invitogen, Thermo Fisher Scientific, Waltham, MA) according to the protocols recommended in the product manual. The expression constructs were then purified using a QIAfilter Plasmid Midi Kit (QIAGEN, Valencia, CA). The TMEM135 expression constructs were transfected into mouse fibroblasts using SuperFect Transfection Reagent (QIAGEN, Valencia, CA) following the manufacture's protocol and cultured for 48 hr.

## Mitochondrial staining

Mitochondria were stained with MitoTracker Red CMXRos (Thermo Fisher Scientific, Waltham, MA) according to the manufacturer's protocol. Briefly, cells were incubated with 200 nM MitoTracker Red CMXRos in cultured medium for 20 min at 37°C. After a rinse with pre-warmed PBS, Cells were ready to be imaged live, or to be fixed with 4% PFA.

## Immunocytochemistry

Cells were cultured on coverslips and were fixed with 4% PFA for 10 min at 4°C. Cells were permeabilized using 0.5% Triton-X in PBS for 30 min followed by blocking in 2% normal donkey serum for 30 min. Then the cells were incubated with primary antibody against TMEM135 (Sigma-Aldrich Cat# SAB2102454, RRID:AB_10611002, St. Louis, MO. This antibody has been discontinued recently from Sigma but the same antibody is available from Aviva Systems Biology Cat# ARP49773_P050, RRID: AB_2048451, San Diego, CA), GFP (Synaptic Systems Cat# 132 002, RRID:AB_887725, Germany) and TOMM20 (Sigma-Aldrich Cat# WH0009804M1, RRID:AB_1843992, St. Louis, MO) in PBS with 0.5% Triton X-100 and 2% normal donkey serum overnight at 4°C. Cells were washed in PBS and then incubated with Alexa 488 conjugated secondary antibody (Thermo Scientific, Rockford, IL) and Cy3 conjugated secondary antibody (Jackson ImmunoResearch Laboratories, West Grove, PA) for 1 hr at room temperature. All immunocytochemistry slides were imaged on a Nikon A1R+ confocal microscope (Nikon Instruments, Melville, NY).

## Western blotting

Cells, mouse brains and retina were homogenized in RIPA buffer (1x PBS with 1% NP-40 and 0.1% SDS, Thermo Scientific, Rockford, IL)) containing a protease and phosphatase inhibitor cocktail (Thermo Scientific, Rockford, IL). Protein concentrations were determined using the Pierce BCA Protein Assay (Thermo Scientific, Rockford, IL) according to the manufacturer's instructions. High- purity mitochondrial fraction was isolated from the whole brain lysate using Qproteome Mitochondria Isolation Kit (QIAGEN, Valencia, CA) according to the manufacturer's protocol. The cell lysate, retinal lysate or the whole brain lysate and isolated mitochondrial fraction containing equal amounts of protein were subjected to SDS–PAGE using 10% Bis-Tris gels and antibodies against HNE (Millipore Cat# 393206-100UL, RRID:AB_211975, Billerica, MA,), β-actin (Cell Signaling Technology Cat# 4970, RRID:AB_2223172, Danvers, MA), TOMM20 (Sigma-Aldrich Cat# WH0009804M1, RRID:AB_1843992, St. Louis, MO), TMEM135 (Sigma-Aldrich Cat# SAB2102454, RRID:AB_10611002, St. Louis, MO. This antibody has been discontinued recently from Sigma but the same antibody is available from Aviva Systems Biology Cat# ARP49773_P050, RRID:AB_2048451, San Diego, CA), SOD1 (Abcam Cat# ab13498, RRID:AB_300402, Cambridge, MA), SOD2 (Abcam Cat# ab13533, RRID:AB_300434, Cambridge, MA), SOD3 (Santa Cruz Biotechnology Cat# sc-32222, RRID:AB_2191977, Dallas, TX), GPx1 (Abcam Cat# ab22604, RRID:AB_2112120, Cambridge, MA), CAT (Novus Cat# NB100-79910, RRID:AB_2071872, Littleton, CO), MFN1 (Abnova Corporation Cat# H00055669-A01, RRID:AB_529865, Taipei City, Taiwan), MFN2 (Abnova Corporation Cat# H00009927-A01, RRID:AB_1204675, Taipei City, Taiwan), LAMP2 (NBP2-22217, Novus Biologicals, Littleton, CO), Lamin B1 (Abcam Cat# ab16048, RRID:AB_10107828, Cambridge, MA), PDI (Cell Signaling Technology Cat# 3501, RRID:AB_2156433, Danvers, MA) and GAPDH (EnCor Biotechnology Cat# MCA-1D4, RRID: AB_2107599, Gainesville, FL). Horseradish peroxidase conjugated secondary antibodies were used (Jackson Immunoresearch, West Grove, PA) prior to detection with a chemiluminescent reagent (Amersham ECL Plus Western blotting detection system, General Electric, Buckinghamshire, UK) and exposure to X-ray film (Thermo Scientific, Rockford, IL). IRDye 800CW or IRDye 680RD secondary antibodies were used (LI-COR Biotechnology, Lincoln, NE) prior to detection with Odyssey CLx imaging system (LI-COR Biotechnology, Lincoln, NE).

## Electron microscopy

Deeply anesthetized mice [12 –month old C57BL/6J (n = 4) and *FUN025* (n = 4)] were used for imaging RPE and inner segment of photoreceptors by electron microscopy. The sample preparation and imaging were performed as previously described (*Johnson et al., 2006*).

## Immuno-electron microscopy

WT fibroblasts were seeded in coated glass coverslips in a 24-well plates overnight. The cells were than transfected with a GFP-TMEM135 vector (pDEST53B6TMEM135) for 48 hr. The cells were fixed with a 4% PFA, 0.1% Glutaraldehyde fixative solution in 0.1 M Phosphate Buffer (PB) for 1 hr at room temperature. Next, the cells were washed with 0.1 M PB (3 × 5 min), incubated with a solution of 0.1% $NaBH_4$ in 0.1 M PB for 10 min, rinsed in 0.1 M PB (4 × 5 min), and permeabilized with a solution of 0.5% Triton-X100 in PBS for 30 min followed by PBS washes (3 × 10 min). The specimens were then transferred to AURION Blocking Solution (Product code 905.002: contains Normal Goat serum, AURION Immuno Gold Reagents & Accessories, Wageningen, The Netherlands) for 30 min, and washed with incubation buffer (0.1% AURION BSA-c, 10 mM $NaN_3$ in PBS at pH 7.4) for 3 × 10 min. Next, the specimens were incubated with anti-GFP antibody (Abcam Cat# ab290, RRID:AB_303395, Cambridge, MA) at 5 µg/ml in incubation buffer at 4°C for overnight. After 6 times washes (10 min each), the specimens were incubated with the secondary antibody Ultra Small gold conjugate reagent [F(ab') 2 Fragment of Goat-anti-Rabbit IgG (H&L), Cat.# 25361, EMS, Hatfield, PA] at 1/100 in incubation buffer overnight. The specimens were then washed with incubation buffer (6 × 10 min) and with PBS (6 × 10 min) followed by post-fixation in 2% glutaraldehyde in 0.1 M PB for 30 min and washes on 0.1 M PB for 2 × 5 min. Next, the specimens were washed with 1X AURION Enhancement Conditioning Solution (ECS) (dilute from 10x concentrated solution, product code 500.055, EMS, Hatfield, PA) for 6 × 5 min and incubated with enhancement mixture (1:20 of DEVELOPER and ENHANCER, EMS, Hatfield, PA) for 2.5 hr in RT in dark. The silver enhancement reaction was stopped by washing the specimens with 1X AURION ECS for 2 × 5 min. The cells were post-fixed with 4% $OsO_4$ for 30 min followed with three times quick washes with 0.1 M PB. The cover-glasses were each transferred into a plastic jar and the cells were dehydrated in a graded series of ethanol. Next, the cells were embedded with 1:1 mix of resin and 100% ethanol, overnight in RT. The samples were then incubated in a 60°C oven for 20 min, and moved to a new plastic jar with fresh 100% plastic resin and incubated for 30 min at 60°C. After that, the samples were moved to another new plastic jar with fresh 100% plastic resin and incubated for another 30 min at 60°C. Next, the samples were polymerized with 100% plastic resin at 60°C for two days. All sectioning was performed on a Reichert Ultracut E (Reichert/Leica Microsystems; Wetzlar, Germany). Ultra-thin sections were cut with a Microstar type SU diamond knife (Microstar Technologies, Huntsville, TX). Ultra-thin sections were stained *en drop* with Uranyl Acetate and Lead Citrate at 60°C. Sections were imaged on a Philips CM120 Scanning Transmission Eletron Microscope (Philips Electron Optics, Eindoven, Netherlands) using Analysis Software (Soft Imaging System Corp., Lakewood, CA).

## Mitochondria isolation

Mitochondria were isolated from the WT mouse brain following the manufacturer's protocol for Mitochondrial isolation kit (MITOISO1, Sigma, Saint Louis, MO). Mitochondria were prepared from four fresh brain tissues following homogenization by an overhead electric motor (~200 rpm). The BSA was added to the extraction buffer to remove lipids. The homogenate was centrifuged at 1000 X $g$ for 5 min. The supernatant liquid was moved to a new tube and centrifuged at 3500 x $g$ for 10 min. Next, the pellet was resuspended in the extraction buffer. The low and high speed centrifugation steps were repeated and the pellet was suspended in the storage buffer (~40 µl per 100 mg tissue).

## Image analysis

For mitochondrial morphometric analysis using ImageJ (RRID:SCR_003070) software, Z-stacks were collected from cells labeled with MitoTracker, and summed projections were generated. Images were thresholded to select mitochondria. From the thresholded fluorescence, binary images were generated, and the size of mitochondria, the number of mitochondria and the total mitochondrial area were measured. Mitochondria number and coverage were generated by dividing the number of mitochondria and total mitochondrial area by the size of cells. Mitochondrial DRP1 puncta were analyzed using NIS-Elements (RRID:SCR_014329, Nikon Instruments, Melville, NY). A binary mask of the mitochondrial channel (MitoTracker) was generated and used it to substrate all extra-mitochondrial DRP1 fluorescence. To select mitochondrial DRP1 puncta for analysis, mitochondrial DRP1 fluorescence was thresholded, and the thresholded image was converted to a binary image. To measure

mitochondrial DRP1 puncta intensity, binary area of mitochondrial DRP1 was divided by binary area of mitochondria. To measure DRP1 puncta fluorescence, puncta identified by thresholding were analyzed for fluorescence intensities.

## Live imaging of TMEM135 and mitochondria

The live imaging of TMEM135 and mitochondria were done in both WT MFs transfected with eGFP-TMEM135 and a stable cell line we generated in HT22 cells expressing GFP-TMEM135. The cells were plated at a density sufficient to reach confluence in two days on glass bottom culture dishes (MatTek Corporation, Ashland, MA) in DMEM with 10% FBS. 30 min before imaging, the cells were live-stained with MitoTracker Red CMXRos (Thermo Fisher Scientific, Waltham, MA), and the culture medium was replaced with DMEM without phenol red (Thermo Fisher Scientific, Waltham, MA) with 10% FBS. Live imaging of TMEM135 and mitochondria was performed on the Revolution XD spinning-disk microscopy system (Andor Technology, Belfast, UK) equipped with the Yokogawa CSU-X1 confocal spinning-disk head; Nikon Eclipse Ti inverted microscope surrounded by an Okolab cage incubator. Andor IQ2 (RRID:SCR_014461) software was used for image acquisition and Imaris ×64 (RRID:SCR_007370, Bitplane AG, Zurich, Switzerland) for image analysis. For live imaging, rapid Z-stacks were acquired using the $100 \times /1.49$ NA Apo TIRF objective (Nikon Instruments, Melville, NY) at 37°C. After background subtraction and smoothing, videos were recorded for WT MFs. For images captured in HT22 cells, Spots algorithm was used to identify TMEM135 spots and follow them through time and cell space, and Surface algorithm was used to identify mitochondria. TMEM135 spots were identified as two groups (mitochondria-bound TMEM135 spots and TMEM135 spots that are not bound to mitochondria) according to their distance to the mitochondria using Spots Close To Surface XTension. One set of TMEM135 Spots (mitochondria-bound TMEM135), located inside the threshold defined region are show in magenta, and another group of TMEM135 spots (TMEM135 not bound to mitochondria), whose shortest distance to the mitochondria surface exceeds the specified threshold are shown in turquoise.

## Mitochondrial membrane potential (MMP) measurements by flow cytometry

$\Delta\Psi m$ was measured using the Mito ID membrane potential detection kit (Enzo Life Sciences, Lorrach, Germany). Cells were seeded overnight and the positive control cells were pretreated with CCCP at a final concentration of 2 uM for 30 min in 37°C, and then harvested by trypsinization, washed, and incubated at room temperature for 15 min with reagent followed by flow cytometry FACSAria II (BD Biosciences, San Jose, CA). Loss of MMP was observed as a decrease in orange and increase in green fluorescence. Results are analysized by FlowJo 7.1 (RRID:SCR_008520, Treestar, Ashland, OR) and presented as a change in ratio of two fluorescence means that correlates to changes in $\Delta\Psi m$.

## Mitochondrial respiration measurements

Mitochondrial respiration was determined using the Seahorse XFe24 Extracellular Flux Analyzer and the XF Cell Mito Stress Test Kit according to the manufacturer's instruction (Agilent Technologies, Santa Clara, CA). Briefly, WT, *FUN025* and Tg-Tmem135 fibroblasts were seeded at $5 \times 10^4$ cells per well in a XF24 cell culture microplate. Cells were washed with XF Assay Media, pre-incubated in a non-$CO_2$ incubator at 37°C for 1 hr in XF Assay Media. For the assay, cells were treated sequentially with 1 μM Oligomycin, 5 μM FCCP (carbonyl cyanide-p-trifluoromethoxyphenylhydrazone), and 1 μM Rotenone plus 1 μM Antimycin A. The Seahorse Wave (RRID:SCR_014526) software is used to design, run and collect the results for all XF assays (all reagents are purchased from Agilent Technologies).

## Analysis of cellular ROS production by flow cytometry

ROS levels were analyzed using Total ROS/Superoxide Detection kit from Enzo Life Sciences, Inc. (Plymouth Meeting, PA). The non-fluorescent, cell-permeable total ROS detection dye was added to cells followed by incubation for 45 min. The dye reacted directly with a wide range of reactive species, such as hydrogen peroxide, peroxynitrite and hydroxyl radicals, yielding a green fluorescent product indicative of cellular production of different ROS/RNS types. The cells were washed twice with PBS in a volume sufficient to cover the cell monolayer and analyzed using a flow cytometry

equipped with standard green filter (490/525 nm or 488 nm laser). Appropriate positive control samples induced with Pyocyanin exhibit bright green fluorescence in the cytoplasm. Cells pre-treated with the ROS inhibitor do not demonstrate any green fluorescence signal upon induction. 1 µM of ROS and superoxide-sensitive fluorescent dyes, and subsequently assayed by flow cytometry FACSAria II (BD Biosciences, San Jose, CA). Double staining with fluorescein isothiocyanate and phycoerythrin was carried out, and data were assessed by FlowJo software version 7.6 (RRID:SCR_008520, Tomy Digital Biology Co., Tokyo, Japan).

## Hyperoxic-exposure

Hyperoxic-exposure of mice was conducted as previously described (*Bozyk et al., 2012*). Briefly, mice were exposed to 75% oxygen for 14 days using a chamber coupled to an oxygen controller and sensor (BioSpherix, Lacona, NY), while the control mice were maintained at normoxia (room air) conditions for the duration of the experiment.

## Apoptosis assay

Terminal deoxynucleotidyl transferase dUTP nicked-end labeling (TUNEL) staining was performed with an Apoptag kit using fluorescein detection (Millipore, Billerica, MA), according to the manufacturer's instructions. Nuclei were counterstained with DAPI, specimens mounted in ProLong Gold antifade reagent (Thermo Fisher Scientific, Waltham, MA). The number of apoptotic cells per retina was counted in three mice retina from three consecutive serial sections and averaged to obtain a mean single TUNEL positive cells value/retina section/mouse.

## Statistical analysis

Sample size was chosen empirically following previous experience in the assessment of experimental variability. No statistical methods were used to predetermine sample size. No animals were excluded. Samples were not randomized and the investigators were not blinded. Statistical analyses were performed in GraphPad Prism 6 (RRID:SCR_002798, GraphPad Software, La Jolla, CA). Significance of the difference between groups was calculated by unpaired Student's two-tailed t test, for experiments comparing two groups, and one-way or two-way analyses of variance (ANOVA) with the Bonferroni-Dunn multiple comparison posttest for experiments comparing three or more groups using $*p<0.05$, $**p<0.01$, $***p<0.001$. $****p<0.0001$. All data are presented as the mean ± the standard error of the mean (s.e.m.) of three or more independent experiments, with three or more replicates per condition per experiment. $P < 0.05$ was considered to be statistically significant.

## Acknowledgements

We thank Satoshi Kinoshita for generation of frozen sections, Drs. Aparna Lakkaraju and Kimberly A Toops for the advice on live imaging experiments, and Drs. David Gamm, Aparna Lakkaraju, and Shigemi Matsuyama as well as Sarah Lewis for comments on the manuscript. The pX330-U6-Chimeric plasmid was a gift from Feng Zhang (Addgene plasmid # 42230). This work was supported by Grants from the National Institutes of Health (NIH R21 EY023061 and R01 EY022086) and a professorship from the Retina Research Foundation (Walter H. Helmerich Research Chair) to AI, NIH grant U01 MH61915 to JST, Core Grant for Vision Research (P30 EY016665) and a Core Grant to Waisman Center (NIH P30 HD003352). JST is an Investigator in the Howard Hughes Medical Institute. Support for ELM was partially provided by the NIH predoctoral training program in Genetics (NIH T32 GM007133).

## Additional information

### Competing interests

JST: Reviewing editor, *eLife*. The other authors declare that no competing interests exist.

### Funding

| Funder | Grant reference number | Author |
| --- | --- | --- |
| National Institutes of Health | T32 GM007133 | Erica L Macke |

| Howard Hughes Medical Institute | | Joseph S Takahashi |
|---|---|---|
| National Institutes of Health | U01 MH61915 | Joseph S Takahashi |
| National Institutes of Health | R21 EY023061 | Akihiro Ikeda |
| Retina Research Foundation | Professorship | Akihiro Ikeda |
| National Institutes of Health | P30 845 EY016665 | Akihiro Ikeda |
| National Institutes of Health | P30 HD003352 | Akihiro Ikeda |
| National Institutes of Health | R01 EY022086 | Akihiro Ikeda |

The funders had no role in study design, data collection and interpretation, or the decision to submit the work for publication.

## Author contributions

W-HL, Performed experiments, Analyzed data, Wrote the manuscript, Conception and design; HH, Perforned experiments, Analyzed data, Aenerated animals and wrote the manuscript; SI, Performed experiments, Analyzed data, Generated animals, Wrote the manuscript; ELM, Analyzed data, Wrote the manuscript; TT, Performed experiments and generated animals; BRP, CL, Performed experiments, Analyzed data; L-FC, Performed experiments; SMS, Generated animals; KJK, CDR, Performed experiments, Generated animals; RFK, JAT, Provided essential equipment and protocol, Advised on experiments, Conception and design; RFM, Performed experiments, Contributed to study design, manuscript writing and preparation; JST, LHP, Generated animals, Contributed to study design, manuscript writing and preparation; AI, Wrote the manuscript and conceived, Directed the study

## Author ORCIDs

Wei-Hua Lee, http://orcid.org/0000-0002-8032-0279
Joseph S Takahashi, http://orcid.org/0000-0003-0384-8878
Akihiro Ikeda, http://orcid.org/0000-0001-8440-3891

## Ethics

Animal experimentation: All experiments were performed in accordance with the National Institute of Health Guide for the Care and Use of Laboratory Animals and were approved by the Animal Care and Use Committee (IACUC) protocols (M01771) at the University of Wisconsin-Madison.

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
