## [Decision Letter]

Thank you for submitting your article "Mouse *Tmem135* mutation reveals a mechanism involving mitochondrial dynamics that leads to AMD pathologies" for consideration by *eLife*. Three experts reviewed your manuscript, and their assessments, together with my own, form the basis of this letter. As you will see, we all thought that the manuscript reports interesting observations that (1) reveal the phenotype of TMEM135 mutation using state-of-the-art mouse genetics and (2) relate mitochondrial dynamics to retinal health.

Regarding revisions, we think that the combination of (1) mutation discovery (forward genetics and recreation with CRISPR of the point mutation), (2) a thorough description of the retinal stress and degeneration phenotype, and (3) a rigorous analysis of the role of TMEM135 in mitochondrial dynamics constitutes a full body of work. The first two categories are quite complete, but reviewer #2 has a number of excellent suggestions for strengthening the third category. Also, the emphasis on aging and AMD is quite speculative at present. These themes should be relegated to roughly 1-2 paragraphs in the Discussion. I would strongly suggest removing "AMD" from the title.

I am including the three reviews at the end of this letter, as there are a variety of specific and useful suggestions in them.

*Reviewer #1:*

Summary:

The work in this manuscript identifies a point mutation in the gene of *Tmem135*, (a previously characterized mitochondrial-associated membrane protein) as the causative agent of pathologies observed in an ENU-induced mutant mouse line (*FUN025*). After advancing the mitochondrial localization, the authors perform a series of experiments to test the function of TMEM135, and report evidence from cultured cells that mitochondrial fission is relatively diminished and that cells with mitochondria having the mutant protein generate elevated reactive oxygen species and superoxides.

Focusing on retinal pathology, and report several phenotypic observations interpreted as a diminished ability to cope with mitochondrial stress.

As the retinal phenotypes of the *Tmem135^FUN025/FUN025^* resemble ones that occur in some strains of aging mice, and further have resemblance to cellular phenotypes (loss of photoreceptor cells, ectopic OPL synapses, GFAP upregulation, reduced ERGs, increased fundus autofluorescence in microglia-like cells) associated with human AMD, the authors propose the mouse as a potentially useful model of AMD.

Critique:

The experimental work in the paper appears carefully done, with adequate documentation and statistics. The characterization of the mitochondrial localization, fission defect, increased ROS and superoxides appear to be genuine advances in the understanding the location and function(s) of TMEM135 in mitochondria. Nonetheless, these results are descriptive in character, and do not establish the biochemical mechanism of TMEM135 function.

The relatively weak nature of the conclusions is seen in such statements:

"Taken together, these results indicate TMEM135 is involved in the regulation of balance between mitochondrial fission and fusion"

"These results suggest the notion that TMEM135 may be required for DRP1 activation"

While the retinal pathologies are not uninteresting, it is highly premature to propose the *Tmem135^FUN025/FUN025^* mouse as a model for AMD. (If mutations in the gene were associated with say, early AMD, or if the function of the TMEM135 were established to be diminished during aging in the human eye, the reviewer would offer a more positive evaluation of this mouse as a model of AMD.)

Overall, the paper doesn't hold together very well. The core problem is that it is very difficult to obtain definitive mechanistic insight into the function of an orphan gene from a single spontaneous (and somewhat complicated) mutation. The authors make the case (Figure 3) that the *Tmem135^FUN025^* mutation creates a protein lacking two transmembrane segments and which flips the C-terminus of the alternatively spliced gene product to the opposite side of the membrane. Consequences would not only include loss of function, but also loss of proper expression level, mistrafficking and mislocalization, and gain of function by the mutant protein product. None of these complications are probed in any depth.

The characterized defects (defective mitochondrial fission, increased ROS and superoxides) simply don't tell one what the protein normally does, and one is stuck with inferences like "it's important for […]"

Bottom line: the work advances understanding of the phenotypic consequences of the specific *Tmem135^FUN025/FUN025^* mutation, both at the level of mitochondrial function and retinal health in mice. However, I don't think novel any definitive insight into the normal role of the TMEM135 has been achieved (because of the complications listed above), and it is certainly premature to argue that this mouse is a good model for AMD.

*Reviewer #2:*

This interesting study characterized a mutant mouse line, *FUN025*, which was created by ENU mutagenesis. These mutant mice have retinal abnormalities, including progressive degeneration of photoreceptor cells. Using genetic mapping and DNA sequencing, the authors found a mutation in *Tmem135* of *FUN025* mice. This mutation caused exon 12 to be skipped and a frame-shift occurred. When the authors introduced the same mutation into wildtype mice using the CRISPR/Cas9 system and crossed this newly created TMEM135 mutant mouse to *FUN025* mice, no complementation for the retinal phenotypes occurred. This suggested that the mutation was responsible for the eye phenotypes.

TMEM135 is predicted to be a polytopic membrane protein. Although a significant portion of TMEM135 was not associated with mitochondria, a portion of TMEM135 formed foci on mitochondria. Fibroblasts isolated from *FUN025* mice showed elongated mitochondria and mitochondria were fragmented when TMEM135 was overexpressed. Live cell imaging showed that TMEM135 was located at the site of mitochondrial fission. However, the majority of mitochondrial TMEM135 foci do not co-localize with Drp1, a protein that mediates mitochondrial fission. Overall, this work is exciting and the identification of mutated *Tmem135* in *FUN025* mice appears to be solid. Although the exact function of TMEM135 is currently unclear, this study discovered a potential new regulator of mitochondrial dynamics and will likely lead to future mechanistic studies. My specific comments are described below.

1) The role of TMEM135 in mitochondrial structure and dynamics was only analyzed in fibroblasts. In order to support the authors' main conclusion that elongation of mitochondria caused retinal degeneration in *FUN025* mice, retinal mitochondrial morphology should be analyzed.

2) In Figure 4, the authors should show the specificity of the anti-TMEM135 antibody using fibroblasts or neurons isolated from *FUN025* mice as negative controls. Similarly, retinal sections of *FUN025* mice should be included. In Figure 4, Tom20 is used as a mitochondrial marker, but the staining pattern does not look like mitochondria. Do the authors have evidence that the anti-Tom20 antibody stains mitochondria in their tissue sections?

3) The authors showed that the mitochondrial fractions contained TMEM135 by Western blotting as evidence of its mitochondrial localization. However, the method was not described and it was unclear if proteins from other organelles were present in this fraction. Proteins specifically localized to other organelles should be tested as controls.

4) The authors described that TMEM135 was associated with lysosome-like organelles solely based on an electron micrograph. This was not convincing. Immunofluorescence micrographs of fibroblasts stained with antibodies that recognize TMEM135 and lysosomal proteins would address this issue.

5) Because elongated mitochondria can be caused by increased mitochondrial fusion or decreased fission, the authors tested whether the overexpression of a mitochondrial fusion protein, Mfn2, could reverse the mitochondrial morphology. They found that Mfn2 overexpression resulted in elongated mitochondria in the TMEM135-overexpressing cells and concluded that mitochondrial fusion was not impaired in these cells. However, it was equally possible that Mfn2 was downregulated and its overexpression overcame such defects in the TMEM135-overexpressing cells. The levels of Mfn2 and other mitochondrial fusion proteins, such as Mfn1 and Opa1, should be measured.

6) The authors stated that *FUN025* mice expressed a C-terminally truncated TMEM135 protein with a reversed topology using a prediction program. However, the mutant protein may be degraded due to compromised folding. Because the anti-TMEM135 antibody recognizes the N-terminus, the authors should test the prediction by Western blotting. Also please confirm the overexpression of TMEM135 by Western blotting.

7) Scale bars are missing.

*Reviewer #3:*

The authors describe the discovery of an ENU mutant mouse with early onset of retinal abnormalities that appear in older WT mice. These changes are associated with abnormal mitochondrial function and higher levels of oxidative stress. The mouse work is thorough and involves the generation of a point mutant used to cement the complementation assay as well as a transgenic overexpressing the protein involved (*Tmem135*). These in vivo assays are supported by in vitro analysis of mitochondrial function and localization of WT and mutant *Tmem135*.

The mouse phenotype is unlike AMD. Although the authors note that the *FUN025* RPE is thicker than normal and has elevated autofluorescence, these are not uniquely associated with AMD. In addition, there are several important distinctions. (1) Dry AMD is characterized by geographic atrophy – that does not seem to pertain here. (2) This mouse model is characterized by panretinal photoreceptor degeneration at levels that are not typical of AMD. (3) One of the early abnormalities noted in *FUN025* is the presence of ectopic neurites. It is not clear that these are encountered in typical AMD. While it would be appropriate to include a paragraph at the end of the Discussion noting that this mouse model possesses some characteristics of the AMD retina, having AMD be front and center from start to finish seems unwarranted.

---

## [Author Response]

*[…] Regarding revisions, we think that the combination of (1) mutation discovery (forward genetics and recreation with CRISPR of the point mutation), (2) a thorough description of the retinal stress and degeneration phenotype, and (3) a rigorous analysis of the role of TMEM135 in mitochondrial dynamics constitutes a full body of work. The first two categories are quite complete, but reviewer #2 has a number of excellent suggestions for strengthening the third category. Also, the emphasis on aging and AMD is quite speculative at present. These themes should be relegated to roughly 1-2 paragraphs in the Discussion. I would strongly suggest removing "AMD" from the title.*

*I am including the three reviews at the end of this letter, as there are a variety of specific and useful suggestions in them.*

*Reviewer #1:*

*[…] Critique:*

*The experimental work in the paper appears carefully done, with adequate documentation and statistics. The characterization of the mitochondrial localization, fission defect, increased ROS and superoxides appear to be genuine advances in the understanding the location and function(s) of TMEM135 in mitochondria. Nonetheless, these results are descriptive in character, and do not establish the biochemical mechanism of TMEM135 function.*

*The relatively weak nature of the conclusions is seen in such statements:*

*"Taken together, these results indicate TMEM135 is involved in the regulation of balance between mitochondrial fission and fusion"*

*"These results suggest the notion that TMEM135 may be required for DRP1 activation"*

We agree that the biochemical function of TMEM135 has not been established from our work. However, we were able to establish that the mutation in this gene is indeed responsible for the retinal pathologies observed in the mouse model. We believe that the association of the novel regulator of mitochondrial dynamics, TMEM135, with retinal health and pathologies related to aging would not have been discovered without this mouse forward genetics study. We have pointed this out in the first paragraph of the Discussion. While further investigation on TMEM135 is necessary, we hope that this study provides an essential step toward understanding the function of TMEM135 in the retina.

*While the retinal pathologies are not uninteresting, it is highly premature to propose the Tmem135^FUN025/FUN025^ mouse as a model for AMD. (If mutations in the gene were associated with say, early AMD, or if the function of the TMEM135 were established to be diminished during aging in the human eye, the reviewer would offer a more positive evaluation of this mouse as a model of AMD.)*

We agree with the reviewer’s comment. We have removed AMD from the title, no longer suggest *FUN025* mice as a model for AMD, and only left AMD as an age-dependent retinal disease with which these mice share some pathologies.

*Reviewer #2:*

*[…] My specific comments are described below.*

*1) The role of TMEM135 in mitochondrial structure and dynamics was only analyzed in fibroblasts. In order to support the authors' main conclusion that elongation of mitochondria caused retinal degeneration in FUN025 mice, retinal mitochondrial morphology should be analyzed.*

We have analyzed the mitochondrial morphology by electron microscopic analysis of the *FUN025* retina. By this analysis, we observed enlarged mitochondria in *FUN025* mice compared to wild-type mice, which has been added to the manuscript (Figure 5).

*2) In Figure 4, the authors should show the specificity of the anti-TMEM135 antibody using fibroblasts or neurons isolated from FUN025 mice as negative controls. Similarly, retinal sections of FUN025 mice should be included. In Figure 4, Tom20 is used as a mitochondrial marker, but the staining pattern does not look like mitochondria. Do the authors have evidence that the anti-Tom20 antibody stains mitochondria in their tissue sections?*

The antigen for the anti-TMEM135 antibody is in the N-terminus of TMEM135, and therefore, TMEM135 with the *FUN025* mutation (with the C-terminus affected) is still recognized by this antibody. For this reason, cells isolated from *FUN025* mice do not serve as negative controls. Nonetheless, we felt it would be informative to include tissues from *FUN025* mice to this analysis and revised the manuscript with those data (Figure 4).

Although the staining with the anti-TOMM20 antibody looks rather diffused, the strong signals are in the outer part of the inner segments of photoreceptor cells where mitochondria are preferentially localized (Stone J et al. Brain Res. 2008 Jan 16;1189:58-69.). Additionally, the same anti TOMM20 antibody has been used in the retina in other publications (e.g. Kim HT et al. Cell Rep. 2015.13(5):990-1002.), which show similar staining patterns as ours. In order to confirm the staining pattern, we have also used other mitochondrial markers on the retinal sections, which produced similar staining patterns as anti-TOMM20. Since the TOMM20 staining in the retina does not clearly show mitochondrial shapes, we have also cultured RPE cells from the mouse retina and stained them with the mitochondrial marker and the TMEM135 antibody. We have added the data which show localization of TMEM135 on mitochondria in RPE cells (Figure 4).

*3) The authors showed that the mitochondrial fractions contained TMEM135 by Western blotting as evidence of its mitochondrial localization. However, the method was not described and it was unclear if proteins from other organelles were present in this fraction. Proteins specifically localized to other organelles should be tested as controls.*

We have added the description for mitochondrial fractionation in the Methods and also added data for proteins specifically localized to the lysosomes, nuclei, and endoplasmic reticulum as controls (Figure 4).

*4) The authors described that TMEM135 was associated with lysosome-like organelles solely based on an electron micrograph. This was not convincing. Immunofluorescence micrographs of fibroblasts stained with antibodies that recognize TMEM135 and lysosomal proteins would address this issue.*

While we have observed localization of TMEM135 on lysosomes by immunofluorescence as suggested by the reviewer, this still does not really prove whether the organelle with TMEM135 that we observed in the EM analysis was the lysosome. We feel that the original statement was not convincing nor appropriate, and have decided to remove it from the manuscript (Figure 4).

*5) Because elongated mitochondria can be caused by increased mitochondrial fusion or decreased fission, the authors tested whether the overexpression of a mitochondrial fusion protein, Mfn2, could reverse the mitochondrial morphology. They found that Mfn2 overexpression resulted in elongated mitochondria in the TMEM135-overexpressing cells and concluded that mitochondrial fusion was not impaired in these cells. However, it was equally possible that Mfn2 was downregulated and its overexpression overcame such defects in the TMEM135-overexpressing cells. The levels of Mfn2 and other mitochondrial fusion proteins, such as Mfn1 and Opa1, should be measured.*

We agree that the scenario described by the reviewer is possible. We have performed Western blotting analysis for OPA1, MFN1, and MFN2 in Tg-Tmem135 fibroblasts. The protein levels of mitochondrial fusion proteins, OPA1, MFN1 and MFN2 were not changed in Tg-TMEM135 MFs compared with WT MFs (Figure 5—figure supplement 2), indicating that over-fragmented mitochondria was not due to downregulation of these mitochondrial fusion factors. Additionally, the levels of these proteins are either unchanged (OPA1 and MFN2) or decreased (MFN1) in *FUN025* MFs compared with WT MFs (Figure 5—figure supplement 2), indicating that the overly fused mitochondrial network observed in *FUN025* MFs was not caused by up-regulation of these mitochondrial fusion proteins.

*6) The authors stated that FUN025 mice expressed a C-terminally truncated TMEM135 protein with a reversed topology using a prediction program. However, the mutant protein may be degraded due to compromised folding. Because the anti-TMEM135 antibody recognizes the N-terminus, the authors should test the prediction by Western blotting. Also please confirm the overexpression of TMEM135 by Western blotting.*

We have performed Western blotting for TMEM135 on brain lysates from *FUN025* mutant, wild- type, and Tg-Tmem135 mice. The data show that TMEM135 is not degraded in *FUN025* mutants, and that the protein expression level is rather higher compared to wild-type mice, which may indicate the existence of a feedback loop from the loss-of-function mutant protein (Figure 3). We have also shown that the TMEM135 protein expression level is higher in Tg-Tmem135 mice compared to wild-type mice (Figure 5—figure supplement 1).

*7) Scale bars are missing.*

We have added scale bars in the figures.

*Reviewer #3:*

*[…] The mouse phenotype is unlike AMD. Although the authors note that the FUN025 RPE is thicker than normal and has elevated autofluorescence, these are not uniquely associated with AMD. In addition, there are several important distinctions. (1) Dry AMD is characterized by geographic atrophy – that does not seem to pertain here. (2) This mouse model is characterized by panretinal photoreceptor degeneration at levels that are not typical of AMD. (3) One of the early abnormalities noted in FUN025 is the presence of ectopic neurites. It is not clear that these are encountered in typical AMD. While it would be appropriate to include a paragraph at the end of the Discussion noting that this mouse model possesses some characteristics of the AMD retina, having AMD be front and center from start to finish seems unwarranted.*

We agree with the reviewer’s comment. We have changed the manuscript so that we only note that this model possesses some pathologies observed in the AMD retina, and have removed it from the title.